

# 1 Aerosol optical properties derived from POLDER-3/PARASOL (2005-2013) over the western Mediterranean Sea – Part 2 : Spatial distribution and temporal variability

Isabelle Chiapello[1], Paola Formenti[2], Lydie Mbemba Kabuiku[2], Fabrice Ducos[1], Didier
Tanré[1], François Dulac[3]
[1]Univ. Lille, CNRS, UMR 8518 – LOA – Laboratoire d'Optique Atmosphérique, F-59000 Lille, France
[2]LISA, CNRS UMR 7583, Université Paris Est Créteil, Université de Paris, IPSL, Créteil, France
[3]LSCE/IPSL, CEA-CNRS-UVSQ, Université Paris-Saclay, Gif-sur-Yvette, France
*Correspondence to:* Isabelle Chiapello (isabelle.chiapello@univ-lille.fr)
**Abstract.** The Mediterranean atmosphere is impacted by a variety of natural and anthropogenic aerosols, which
exert a complex and variable pressure on the regional climate and air quality. In this study, we investigate aerosol
spatial distribution and temporal evolution over the western Mediterranean Sea (west of longitude 20°E) using the
full POLDER-3/PARASOL aerosol data record derived from the operational clear-sky ocean algorithm (collection
3) available from March 2005 to October 2013. This 8.5-yr satellite data set includes retrievals at 865 nm of the
total, fine, and coarse mode aerosol optical depth (AOD, $AOD_F$, and $AOD_C$, respectively), Angström exponent
(AE), and the spherical/non-spherical partition of the coarse-mode AOD ($AOD_{CS}$ and $AOD_{CNS}$, respectively). In a
previous paper (Formenti et al., 2018), these POLDER-3-derived aerosol properties have been carefully validated
over the study region, based on coincident ground-based and airborne aerosol measurements. Here we analyze the
spatial distribution, the seasonal cycle and interannual variability of this ensemble of products in three latitude
bands (34-38°N, 38-42°N, and >42°N) and for three sites (Ersa, Barcelona, Lampedusa) distributed on the western
basin Overall the POLDER-3 AOD spatial distribution exhibits a well-known south-to-north decreasing gradient,
and a seasonal cycle characterized by enhanced aerosol loads in spring and summer, both controlled by Saharan
dust. POLDER-3 retrievals of AE, $AOD_F$, $AOD_C$, and fine mode fraction ($AOD_F$/AOD) highlight the influence of
coarse particles in the southern part of the region, off the north African coast, and higher relative contribution of
fine particles in the northern part, off the south European coast, with all year long persistent elevated loads over
the Adriatic Sea. Over the rest of the western Mediterranean Sea, POLDER-3 retrievals show a more homogeneous
spatial distribution of fine particles than that of coarse particles, even though climatological means of $AOD_F$
highlight seasonal differences in the order of a factor 2 between the cleanest conditions occurring in the southern
part of the basin in winter and those most polluted observed in its northern part in Spring. The seasonal and spatial
variability of $AOD_{CNS}$ is close to that observed for $AOD_C$, whereas POLDER-3 exhibit relatively low and weakly
variable levels of coarse spherical particles ($AOD_{CS}$ < 0.05). Over the whole 2005-2013 period, annual POLDER-
3 AOD evolution shows a decreasing trend (≥ 0.003 per year in absolute value). Such a decrease is much more
pronounced for $AOD_F$ (≥ 0.002 per year) than for $AOD_C$ (≤ 0.002 per year). Our analysis also suggests that the
North Atlantic Oscillation (NAO) index explains a significant part of the interannual variability of POLDER-3
$AOD_C$, reflecting its role on the frequency of Saharan dust transport over the region. Finally, the POLDER-3
dataset highlights an improvement of air quality related to the fine aerosol component, with an evolution toward



more frequent occurrence of clean conditions ($\geq$ 70% of daily $AOD_{F\text{-}865\,nm}$ < 0.05) at the end of the period of study
(2010-2013) over the western Mediterranean Sea.

**1 Introduction**

Due to the contributions of diverse natural and anthropogenic sources and because of their relatively short lifetime
in the troposphere, aerosols consist in a complex, timely and spatially variable mixture of particles (Boucher,
2015). As aerosol impacts, especially in terms of air quality degradation and radiative forcing contribution to
climate change, strongly depend on both very variable aerosol loads and properties, they require a dedicated
reliable monitoring. Despite a number of measurements efforts deployed in the last decades (Laj et al., 2009;
Pandolfi et al., 2018; Formenti, 2020; Laj et al., 2020), the variety of atmospheric particles, in terms of loads, size
ranges, shapes, chemical compositions, and optical properties remains partially characterized. Indeed, the
monitoring of the spatial, temporal, and vertical variability of all these physico-chemical parameters in both an
accurate and comprehensive way is still a challenge. Significant advances have been achieved by intensive field
experiments deploying detailed but limited in time and space *in situ* measurements of aerosol chemical, physical,
and optical properties (e.g. Denjean et al., 2016; Di Biagio et al., 2016). In parallel, remote sensing observations,
especially those from ground-based global aerosol networks, like AERONET (Holben et al., 2001), and dedicated
advanced aerosol satellite sensors, like MODIS (MODerate resolution Imaging Spectrometer) or POLDER
(POLarization and Directionality of the Earth's Reflectances) (Tanré et al., 2011; Bréon et al., 2011; Remer et al.,
2020), have made considerable progress in expanding in time and space the aerosol datasets acquired from field-
experiments. Thus, remote sensing has become an essential complementary tool, able to provide unique repetitive
and large-scale view of aerosol loads and properties evolution. The combination of both types of measurements,
i.e. detailed in situ aerosol characterization and long-term repetitive aerosol properties monitored by space-borne
sensors is required to improve current understanding of their evolution in terms of loads and properties and to
reduce uncertainties on their impacts.
This paper is dedicated to a regional aerosol analysis based on retrievals from POLDER-3/PARASOL
(Polarization & Anisotropy of Reflectances for Atmospheric Sciences coupled with Observations from a Lidar)
satellite sensor over the period 2005-2013 in the western Mediterranean Sea. This region, impacted by
demographic pressure and air quality degradation, is under the influence of both anthropogenic and natural
aerosols, emitted from different types of continental and marine sources (e.g. Lelieveld et al., 2002; Di Biagio et
al., 2015; Ancellet et al., 2016; Chazette et al., 2016, Claeys et al., 2017; Michoud et al., 2017; Chazette et al.,
2019). Therefore, in the recent years, it has experienced an increasing scientific interest as shown by a number of
studies dedicated to Mediterranean aerosol characterization through large-scale field-experiments (e.g., Di Biagio
et al., 2015; Mallet et al., 2016; Ricaud et al., 2018 and references therein), modeling efforts (Rea et al., 2015;
Menut et al., 2016, Sič et al., 2016; Chrit et al., 2018; Drugé et al., 2019), and satellite observations analysis (Nabat
et al., 2013; Floutsi et al., 2016). Historical long-term aerosol satellite datasets have been used to investigate the
influence and evolution of north African mineral dust transported over this region (Dulac et al., 1992; Moulin et
al., 1998; Antoine and Nobileau 2006; Gkikas et al., 2013, 2016). Specific studies, often based on MODIS aerosol
retrievals or combining MODIS to other complementary aerosol satellite data sets, have attempted to separate the
contributions of different aerosol types prevailing in the Mediterranean region, i.e. maritime aerosols,
continental/anthropogenic aerosols, and African dust (Barnaba and Gobi, 2004; Hatzianastassiou et al., 2009;



Georgoulias et al., 2016). Despite the increasing number of satellite-based aerosol studies, especially in the East
part of the Mediterranean area (Georgoulias et al., 2016; Shaheen et al., 2020), it is noticeable that no investigation
of POLDER-3/PARASOL aerosol products (Herman et al., 2005; Tanré et al., 2011) has been performed yet over
this region, despite its potential for monitoring the size-resolved aerosol properties over sea surfaces over its almost
9 years period of operation (Formenti et al., 2018).
In order to ensure a reliable regional view of aerosol loads and properties evolution from satellites, aerosol
retrievals derived from different sensors and algorithms require careful evaluation. POLDER-3 aerosol retrievals
validation has been performed for derived total and fine aerosol optical depth (AOD) through statistical
comparison to sun/sky photometer data of the AERONET network at a global scale (Bréon et al., 2011). In a first
dedicated paper (part 1 of the present paper: Formenti et al., 2018), we lead a regional comprehensive quality
assessment of POLDER-3 derived aerosol parameters over the western Mediterranean Sea, based on both aerosol
measurements from 17 ground-based coastal and insular AERONET sites over the period 2005-2013, and in situ
airborne observations available during summer 2012 and 2013 Chemistry-Aerosol Mediterranean Experiment
(ChArMEx) experiments (Di Biagio et al., 2015; Mallet et al., 2016). Our analysis has highlighted quality and
robustness of POLDER-3 operational aerosol retrievals over oceans, especially total, fine, and coarse AOD (AOD,
$AOD_F$, and $AOD_C$) at 865 nm, Angström Exponent (AE), and the spherical and non-spherical partition of coarse-
mode AOD ($AOD_{CS}$ and $AOD_{CNS}$) over this region.
In this paper, the advanced aerosol data set provided by POLDER-3 over its operating period, i.e. from March
2005 to October 2013, is investigated in terms of spatial variability and temporal evolution of aerosol load and
properties over the western Mediterranean Sea.
**2 POLDER-3 instrument and derived aerosol operational products over ocean**
POLDER-3 (POLarization and Directionality of the Earth's Reflectances) instrument on board the PARASOL
(Polarization & Anisotropy of Reflectances for Atmospheric Sciences coupled with Observations from a Lidar)
mission is dedicated to advanced aerosol monitoring (Tanré et al., 2011). PARASOL, launched in December 2004
in order to be part of the A-Train, has been in operation from March 4, 2005 to October 10, 2013. Over this period,
data availability is 91%. The explanations for the 9% loss of data are multiple: orbital maneuvers, instrument put
on standby for security reasons, data transmission between the payload and the receiving station, and problems
encountered with the stellar sensor. POLDER-3 payload consisted of a digital camera with a 274 x 242 –pixel
CDD detector array, wide-field telecentric optics and a rotating filter wheel enabling measurements in 9 spectral
channels from blue (443 nm) to near-infrared (1020 nm). Polarization measurements were performed at 490 nm,
670 nm, and 865 nm. With an acquisition of a sequence of images every 20 sec, the instrument could observe
ground targets from up to 16 different angles, +/-51° along track and +/-43° across track (Tanré et al., 2011). The
original pixel size is 5.3 km x 6.2 km at nadir. Algorithms have been developed to process the POLDER
measurements in order to retrieve aerosol parameters at 18.5 x 18. 5 km$^2$ superpixel resolution (3 x 3 pixels). In
this paper, we use the operational clear-sky ocean retrieval algorithm (Herman et al., 2005) derived from collection
3, corresponding to the latest update performed in 2014 that included calibration improvements (Fougnie, 2016).
This algorithm, described in details by Herman et al. (2005) and Tanré et al. (2011), has been slightly improved in
collection 3 regarding non-spherical particles in the coarse mode (Formenti et al., 2018). Briefly, it is based on the
total and polarized radiances measured at 670 and 865 nm. Using a look up table (LUT) built on aerosol


microphysical models (described in Table S1 in the Supplement of Formenti et al., 2018), the algorithm
recalculates for each clear sky pixel the observed polarized radiances at several observational angles. Importantly,
in the aerosol models used for the inversion, aerosols are considered as non-absorbing (the imagery part of the
refractive index is assumed as zero) and the real part of their refractive index is invariant between 670 and 865
nm. The aerosol number size distribution is lognormal and bimodal with an effective diameter smaller (larger)
than 1.0 µm for the fine (coarse) mode. The coarse mode includes a non-spherical fraction based on the spheroidal
model from Dubovik et al. (2006), whereas a Mie model for homogeneous spherical particles is used to calculate
multi-spectral and multi-angle polarized radiances. As an improvement compared to former versions of the
algorithm, the effective diameter of the spheroidal model is allowed to take two values (namely 2.96 and 4.92 µm)
in collection 3 (Table S1 of Formenti et al, 2018). Within the coarse mode, the non-spherical fraction is set to 5
discrete values (0.00, 0.25, 0.50, 0.75, and 1.00, Tanré et al. (2011)). A quality flag index (0 indicating the lowest
and 1 the highest quality) is attributed to each superpixel depending on the inversion quality. As in Formenti et al.
(2018), only POLDER-3 aerosol products derived from pixels with a quality flag $\geq 0.5$ have been considered in
our analysis. In the present study, we focus on the western Mediterranean region, west of longitude 20°E,
considering the main aerosol parameters derived by POLDER-3 ocean operational algorithm: (i) available for all
clear sky pixels: total, fine, and coarse aerosol optical depth (respectively AOD, $AOD_F$, and $AOD_C$) at 865 nm,
and Angström Exponent between 670 and 865 nm (AE), (ii) available only when the geometrical conditions are
optimal (scattering angle range of roughly 90°-160°): spherical and non-spherical fractions of the AOD in the
coarse mode ($f_{CS}$ and $f_{CNS}$ respectively), allowing to assess $AOD_{CS}$ and $AOD_{CNS}$ (spherical and non-spherical
coarse AOD, respectively) at 865 nm. The quality of these POLDER-3 derived aerosol parameters has been
evaluated over the region of interest by Formenti et al. (2018), using co-located in situ airborne measurements
from summer 2012 and 2013 field-experiments and coincident ground-based AERONET data available from 17
insular and coastal sites over the whole POLDER-3 operation period (2005-2013). This first comprehensive
regional evaluation has provided new assessments of uncertainties and highlighted the good quality of collection
3 POLDER-3 aerosol data set over our area of interest (Table 4 of Formenti et al., 2018). In our regional analysis
of spatial distribution and temporal variability of POLDER-3 aerosol retrievals, the AOD, $AOD_F$, and $AOD_C$
derived at 865 nm will be complemented, through an extrapolation with the Angström Exponent, by those at 550
nm, which is the standard wavelength of many aerosol satellite retrievals and model simulations (Nabat et al.,

151    2013).

**3  Results**
**3.1  Mean regional and seasonal picture (2005-2013)**
The climatological (March 2005 – October 2013) seasonal maps of POLDER-3 derived AOD, AE, $AOD_F$, $AOD_C$,
$AOD_F$/AOD (i.e. Fine Mode Fraction or FMF), $AOD_{CNS}$, and $AOD_{CS}$ at 865 nm over marine areas in the region
30-50°N, 10°W-20°E, i.e. mainly the western Mediterranean Sea, are shown in Figure 1. The total AOD (left
panels) exhibits a pronounced seasonality with minimum values in winter (defined by the December-January-
February months): AOD < 0.10 over most of the region of study. In spring (March-April-May), AOD shows an
increase, especially intense over the southeastern part of the region between Italy and Africa, whereas the
maximum AOD values ($\geq 0.20$) are reached in summer (June-July-August) over the whole southern part of the
area. In autumn (September-October-November), the AOD over the region are mostly low, comparable to winter



loads, except over the southeastern part of the domain, especially over the Ionian Sea, and off the coast of Tunisia,
Lybia and south of Sicily, where they reach moderate values (range 0.10 – 0.15). This area of enhanced aerosol
transport is geographically similar to that associated to maximum AOD (~ 0.20) in spring. In general, the seasonal
POLDER-3 total AOD maps exhibit a well-established south-to-north gradient, with a decrease of values toward
the northern part, reflecting the high influence of aerosol sources from the North African continent. This aerosol
spatial distribution is consistent with that derived by other satellite sensors over the Mediterranean basin (for
example Moulin et al., 1998, Barnaba and Gobi, 2004; Papadimas et al., 2008). The $AE_{865\text{-}670\ nm}$ seasonal maps
(second column panels) highlight the influence of coarse aerosols (associated with low AE values) in the south
part of the region off the north African coast, and higher contribution of fine particles along the coasts of Europe,
especially over the Adriatic Sea, where AE values are equal or higher than 1, in all seasons. $AOD_F$, $AOD_C$, and
$AOD_F$/AOD (FMF) seasonal maps, shown in the three central column panels, confirm this pattern of spatial
variability, typical of coarse and fine aerosol repartition in the Mediterranean basin. The seasonal and spatial
variability of $AOD_{CNS}$ is close to that observed for $AOD_C$, whereas POLDER-3 retrievals of $AOD_{CS}$ suggest a
relatively homogeneous repartition of coarse spherical particles, with low values ($AOD_{CS} < 0.05$), and no
substantial spatial and seasonal variations (right panels of Figure 1). Figure S1 of the supplementary material
complements these POLDER-3 seasonal maps at 865 nm, with AOD, $AOD_F$, $AOD_C$, and $AOD_F$/AOD (i.e. FMF)
extrapolated at 550 nm. At this wavelength, AOD reach higher values ($\geq 0.30$ during summer maximum), $AOD_F$
are strongly enhanced (values up to 0.16-0.20) compared to 865 nm ($< 0.08$), whereas $AOD_C$ values are only
slightly modified. These ranges of values are consistent with the stronger wavelength dependence of AOD of small
particles, characterized by high AE values, inducing pronounced increase of $AOD_F$ values toward shorter
wavelengths. Thus, the spatial distribution of POLDER-3 $AOD_F$ at 550 nm is characterized by maximum values
($> 0.10$) over the eastern part of the region of study, and seasonal peaks in spring and summer. North of the Adriatic
Sea, POLDER-3 highlights an area characterized by all-year persistent high values of $AOD_F$ ($> 0.12$ at 550 nm),
most probably reflecting accumulation of pollution particles due to influence of regional anthropogenic sources
(as for example from Northern Italy in the Po Valley).

**3.2  Sub-regional features**

In order to examine more deeply the seasonal variations of POLDER-3 aerosol retrievals accounting for the south-
to-north gradient observed in Figure 1, the area of study has been divided into three main latitudinal sub-regions.
These regions are illustrated in Figure 2. They correspond respectively to the northern part (north of latitude 42°N:
zone 1 called NW MED), the central part (latitude band 38 – 42 °N, zone 2 called CW MED), and the southern
part (south of latitude 38°N: zone 3 called SW MED) of the western Mediterranean Sea (6°W- 20°E).
Figure S2 of the supplementary material reports the statistics of the POLDER-3 retrievals over the March 2005-
October 2013 time period in each-sub-region, with mean and standard deviations, maximum and minimum values
of number of available clear-sky superpixels (left column) and number of available days of observations for each
month and year (right column). The maximum number of POLDER-3 superpixels (left panels) is 434 in NW MED
and up to 1232 in SW MED and 1384 in CW MED, reflecting the smaller size of the NW-MED sub-region. As
expected, more POLDER-3 retrievals are available in summer than in winter months, due to the higher influence
of cloudiness during the cold season. The number of days with aerosol retrievals by month and year for each sub-
region (right column) highlights that more than 50% of daily POLDER-3 retrievals are available for most of the



months of the whole time period. A few exceptions occur for some specific months, as July 2007 and July 2010,
common at the three sub-regions due to missing data during these periods related to instrumental problems with
the solar sensor (only 28% and 14% of data available, respectively). A reduced number of days of POLDER-3
aerosol retrievals is observed in the NW MED sub-region in November and December 2012 (respectively 11 and
4 days, Figure S2b), likely due to an unfavorable combination of cloudiness and spatial coverage in the
northernmost part of our study region during these two months. Beyond this particular case, this analysis generally
suggests that the cloudiness significantly reduces the number of POLDER-3 pixels available over each sub-region
from October to March (Figure S2a,c,e), even though the number of available days of POLDER-3 observations
remains reasonable (Figure S2b,d,f).

Figure 3 illustrates the 8- or 9-year climatological mean over March 2005 – October 2013 of monthly POLDER-
3 derived aerosol parameters at 865 nm over the three sub-regions defined in Figure 2. The averaged seasonal
cycle of AOD is relatively similar over the north and central parts of the basin, whereas the southern part shows
generally higher total aerosol loads, and a more pronounced seasonal variability, with two maxima in April-May
and July (mean $AOD_{865\,nm} > 0.15$). The mean monthly variations of the POLDER-3 $AOD_F$ integrated over the
three sub-regions are remarkably similar, in agreement with previous analysis based on ground-based AERONET
observations suggesting that the aerosol fine mode is, to some extent, relatively homogeneously distributed over
the western Mediterranean region (Lyamani et al., 2015; Sicard et al., 2016). Conversely, the north-south gradient
clearly appears for $AOD_C$ (right column middle panel of Figure 3), especially for the SW MED area, consistently
with what is observed for total AOD. The seasonal variations of the monthly-averaged AE (left column middle
panel) reflect the north–south gradient of aerosol sizes, with an increased influence of smaller particles toward the
north, a pattern confirmed by the monthly evolution of FMF (left column, bottom panel). The monthly-averaged
$AOD_{CS}$ (right column, bottom panel) shows very low seasonal and spatial variability, as previously observed in
Figure 1, whereas the POLDER-3 mean $AOD_{CNS}$ seasonal cycle illustrates much more pronounced monthly and
north-south evolution, in coherence with those of $AOD_C$ and total AOD. Figure S3 in the supplementary material
illustrates the climatological mean of monthly POLDER-3 AOD, $AOD_F$, $AOD_C$, and FMF extrapolated at 550 nm,
confirming the patterns displayed Figure 1, especially the marked increase of $AOD_F$ values, and FMF at this
wavelength.
The POLDER-3 mean seasonal aerosol retrievals displayed in Figure 1 and 3 at 865 nm are summarized in Table
1a, those extrapolated at 550 nm (Figures S1 and S3) in Table 1b. The multi-annual averages of AOD, $AOD_C$ and
$AOD_{CNS}$ at 865 nm in Table 1a confirm the north-south gradient with minimum values in the north part (0.090,
0.055, and 0.043 respectively for AOD, $AOD_C$, and $AOD_{CNS}$) compared to the south part of the western
Mediterranean basin (0.124, 0.091, 0.073 respectively). POLDER-3 AE and FMF mean multi-annual values
consistently highlight an increase in the coarse component of AOD toward the south. In terms of multi-annual
averages, the $AOD_F$ remains relatively uniforms, with some minor variations indicating minimum fine mode
aerosol loads in the central area (0.032 in CW MED), maximum in the north (0.035 in NW MED) and intermediate
values in the south part (0.033 in SW MED), these variations being more pronounced at 550 nm (Table 1b).
Seasonal multi-annual averages of $AOD_F$ highlight differences in the order of a factor 2 between minimum values
in the south in winter (around 0.02 at 865 nm, 0.06 at 550 nm) and maxima in spring (around 0.04 at 865 nm, and
0.12 at 550 nm), especially in the northern part of the region. The POLDER-3 derived mean multi-annual $AOD_{CS}$



at 865 nm (Table 1a) reveal some seasonal variability, with maximum values in summer in the south part (0.031)
and minimum in winter in the northern part (0.013). Although reasons for such an evolution are not fully
understood, considering the similarity with that of $AOD_{CNS}$, this variability could be partly related to the influence
of North African dust transport rather than fully representative of a background coarse sea-salt fraction (Claeys et
al., 2017). Indeed, Saharan dust might include a spherical coarse aerosol fraction following mixing with soluble
secondary components such as sulfate and nitrate (Drugé et al., 2019).

**3.3 Temporal evolution at selected sites**

The previous regional analysis is complemented by the investigation of the POLDER-3 aerosol properties around
three contrasted AERONET sites of the western basin: Ersa (43.00367°N, 9.35929°E, altitude 80 m), the
northernmost site located on northern coast of Corsica Island, France; Lampedusa (35.51667°N, 12.63167°E, alt.
45 m) the southernmost site located on the northwestern coast of Lampedusa Island, Italy; Barcelona (41.38925°N,
2.11206°E, alt. 125 m) the westernmost site located in a urban/coastal environment on the shore of northeastern
Spain (Figure 2). Ersa and Barcelona are sites under the influence of long-range Saharan dust transport, whereas
Lampedusa is subject to short to medium-range dust transport. Ersa and Lampedusa are marine background sites
with some anthropogenic influence, Barcelona is located in a heavily polluted environment. Ersa and Lampedusa
were the two super-sites of the ChArMEx (The Chemistry-Aerosol Mediterranean Experiment) collaborative
research program, and Barcelona, which is also part of EARLINET/ACTRIS network, one of the secondary sites
of this program (Mallet et al., 2016). In this context, many experimental set up of in situ aerosol measurements
provided detailed aerosol characterization. Additionally, the long-term AERONET routine aerosol measurements
at these sites have been used for the comprehensive regional validation of POLDER-3 retrievals presented in
Formenti et al. (2018). Here we considered the same POLDER-3 dataset, by selecting superpixels within ±0.5°
around the AERONET sites, corresponding to a maximum number of 17 at Ersa, 28 at Lampedusa, and 13 at
Barcelona.

**3.3.1 Monthly time series**

Figure 4 illustrates the month-to-month evolution from March 2005 to October 2013 of POLDER-3 retrievals at
865 nm, extracted at Ersa, including (a) AOD, (b) $AOD_F$ and $AOD_C$, (c) $AOD_{CNS}$ and $AOD_{CS}$, (d) $AE_{865-670}$ and
FMF. The total aerosol load derived from POLDER-3 at this site show a marked variability, both at seasonal and
interannual time scales, with a maximum recorded value of monthly averaged AOD of 0.21 (June 2007), and
winter minimum values of AOD around 0.05. Monthly-averaged AOD values above 0.10 occur mostly during
spring (April-May) and summer (June-July) seasons. Interestingly, Figure 4a highlights some additional peak of
AOD occurring in autumn for specific years, as for example in September-October 2008. Month-to-month
evolution of AE and FMF reported on Figure 4d show remarkably similar variability, which confirms that the AE
is a good proxy of the proportion of fine particles component relative to total AOD. The average monthly FMF of
the AOD at 865 nm at Ersa is estimated at 37% by POLDER-3 in all clear-sky conditions, with a range of monthly
mean values between 18% and 65%, and only 17 months over 103 (i.e. 17%) with FMF greater or equal to 50%
(considering 865 nm wavelength). These occurrences are mostly recorded in late winter (February-March) and
autumn (September-October-November) and are generally characterized by moderate monthly mean AOD values
(between 0.06 and 0.13). Figure 4b illustrates the month-to-month evolution of $AOD_F$ and $AOD_C$, confirming that



at this wavelength (865 nm), the AOD is mostly dominated by the coarse mode fraction (63% on average). For
some specific months of the time series, as for example September 2005, July 2006, September 2009,
February/March 2010, the POLDER-3 derived $AOD_F$ is higher than $AOD_C$, but these cases are rather rarely
observed in comparison to those corresponding to the dominance of the AOD coarse mode component. The
monthly evolution of POLDER-3 $AOD_{CS}$ and $AOD_{CNS}$ at Ersa, reported in Figure 4c also suggests a strong
domination of non-spherical particles in the coarse mode AOD over most of the 8.5-year time series. A few cases
with $AOD_{CS}$ greater or equal to $AOD_{CNS}$ did occur over the period, as for example in August 2010, June 2011, or
July 2012. They need to be explored, although they clearly do not reflect the most frequent aerosol conditions in
Ersa. Considering only the POLDER-3 retrievals available in Best Viewing Conditions, the averaged repartition
in terms of aerosol size mode and shape contributions to the total AOD at 865 nm at Ersa are 36% for the fine
AOD, 44% for the non-spherical coarse mode and 20% for the spherical coarse mode. These estimates are
consistent with the POLDER-3 data set available in all clear sky conditions, which estimate 63% of coarse mode
AOD versus 37% of fine mode AOD at 865 nm.
Similarly to Figure 4, Figure 5 and 6 illustrate the month-to-month evolution from March 2005 to October 2013
of POLDER-3 retrievals at 865 nm, extracted at Barcelona and Lampedusa respectively, including (a) AOD, (b)
$AOD_F$ and $AOD_C$, (c) $AOD_{CNS}$ and $AOD_{CS}$, (d) $AE_{865-670}$ and FMF. Consistently with the influence of short to
medium range Saharan dust transport expected in Lampedusa, POLDER-3 AOD show their highest monthly mean
values at this site (up to 0.44 in May 2011, Figure 6a), compared to both Ersa (max of 0.21 in June 2007, Figure
4a) and Barcelona (max of 0.24 in June 2006, Figure 5a). These maximum AOD values are clearly associated to
coincident maximum values of monthly mean $AOD_C$, with 0.39 in May 2011 in Lampedusa (Figure 6b), 0.18 in
June 2006 in Barcelona (Figure 5b), and 0.16 in June 2007 in Ersa (Figure 4b). Figure 4-6 clearly highlight that
POLDER-3 monthly mean AOD values above 0.10 are much more frequent in Lampedusa (66% of frequency
over the 104 months of POLDER-3 observations) than in Barcelona (43% of frequency) and Ersa (30%). The
contrast between the three sites is even more pronounced considering the $AOD_C$ retrievals, as the frequency of
monthly values above 0.10 (44%, 22%, and 5% for Lampedusa, Barcelona, and Ersa, respectively) clearly
highlights the more frequent impact of coarse particles, especially non-spherical desert dust, in Lampedusa.
Conversely, the monthly evolution of $AOD_F$ reported in Figure 4b, 5b, and 6b does not show such a marked
contrast, nor with respect to the maximum values, rather comparable at the three sites (0.072, 0.074, and 0.076 in
Ersa, Barcelona, and Lampedusa, respectively), or the frequency of monthly mean values above 0.04 (27%, 31%
and 34% respectively). At Lampedusa, monthly $AOD_F$ are always substantially lower than monthly $AOD_C$ (Figure
6b). At Barcelona monthly $AOD_F$ are mostly below $AOD_c$ (Figure 5b), with the noticeable exception of a few
months over the time period, generally in winter or late summer (Feb. 2006, Feb. 2008, Sept. 2009 for example),
characterized by the dominance of $AOD_F$. These months with POLDER-3 mean derived FMF greater than 50%
represent a frequency of 10% over the whole monthly data set in Barcelona (Figure 5d), and 0% in Lampedusa
(Figure 6d). Compared to their frequency in Ersa (17%, Figure 4d), POLDER-3 retrievals of fine and coarse
components of AOD suggest that the influence of fine particles is more frequent in Ersa, possibly due to the
transport of polluted air masses from highly industrialized regions (Po Valley, Marseille-Fos-Berre for example)
in the north part of the basin (Mallet et al., 2016). These features may also simply reflect the more frequent
influence of desert dust at Lampedusa and in a less extent at Barcelona, which may hide the possible influence of
fine aerosols of anthropogenic origin at these two sites. Over the whole POLDER-3 observing period, maximum



monthly mean values of $AOD_{CS}$ range from 0.058 in Ersa (March 2008, Figure 4c) to 0.075 in Lampedusa (April
2008, Figure 6c) and 0.090 in Barcelona (November 2009, Figure 5c). Frequencies of monthly mean POLDER-3
$AOD_{CS}$ values above 0.03 are 13%, 31%, and 38% at Ersa, Barcelona, and Lampedusa respectively. Such a
variability suggests some impact of desert dust on $AOD_{CS}$, although the contribution of sea-salt particles or a
combination of both aerosol types cannot be excluded without further investigations. Maximum monthly $AOD_{CNS}$
values range from 0.109 at Ersa (Sept. 2008 and May 2009, Figure 4c) to 0.210 at Barcelona (Nov. 2009, Figure
5b) and 0.220 at Lampedusa (March 2005, Figure 6b). Frequencies of monthly mean POLDER-3 $AOD_{CNS}$ values
above 0.03 reach 91% in Lampedusa, 70% in Barcelona, and 67% in Ersa. Considering only the POLDER-3
retrievals available in Best Viewing Conditions, the averaged contributions in terms of aerosol size and shapes at
Barcelona are rather similar to those estimated at Ersa, with 34% of fine AOD, 46% of coarse non-spherical AOD
and 20% of coarse spherical AOD at 865 nm. At Lampedusa, the averaged contribution of fine AOD is reduced to
26%, with a higher contribution of coarse non-spherical AOD (55%), and a rather constant relative contribution
of coarse spherical AOD (19%).
**3.3.2 Daily time series**
Figure 7 shows the daily evolution from March 4, 2005 to October 10, 2013 of POLDER-3 AOD (a), $AOD_F$(b),
$AOD_C$(c), $AOD_{CS}$, and $AOD_{CNS}$ (d) at 865 nm at Ersa, Barcelona, and Lampedusa. Table 2 presents a statistical
summary of the daily POLDER-3 aerosol retrievals for these three sites. The range of AOD values varies from
0.01 to 0.68 at Ersa, 0.01 to 1.05 at Barcelona, and 0.02 to 4.72 at Lampedusa, indicating the occurrence of extreme
AOD events at the southernmost site of Lampedusa. Daily AOD > 0.3 occur 9% of the time in Lampedusa. For
these days, POLDER-3 retrieves low AE (0.41 in average), low FMF (21% in average), and high contribution of
non-spherical aerosol fraction in the coarse mode (87% in average), consistently with the dominant influence of
desert dust. Such an influence is verified at Barcelona as well: although much less frequent (less than 3% of
occurrence), daily AOD >0.3 are associated to average AE values of 0.44, average FMF of 21% (maximum 45%)
and average non-spherical aerosol fraction in the coarse mode of 85%. Finally, at Ersa POLDER-3 retrievals of
daily AOD > 0.3 are rare (< 2% of occurrence over the POLDER-3 observing period), and characterized by the
same properties typical of desert dust influence (mean AE 0.31, mean FMF 16%, mean non-spherical aerosol
fraction in the coarse mode of 71%). These POLDER-3 retrievals are consistent with the Gkikas et al. (2013)
climatology of intense desert dust events in the Mediterranean, which recorded extreme dust episodes mostly in
the southern part of central Mediterranean, where Lampedusa is located. These authors also reported that these
extreme desert dust episodes are characterized by $AOD_{550nm}$ values > 2.5 and up to 4. At Lampedusa, Figure 7b
and 7c suggest that the most intense AOD peaks are always associated to an increase in $AOD_C$, and in some cases
in $AOD_F$ as well. Thus, for the three main sites considered here, the aerosol retrievals recorded by POLDER-3
from 2005 to 2013 highlight the high variability of both the total and the size- and shape-segregated components
of the AOD. The background aerosol conditions, corresponding to low POLDER-3 $AOD_{865 nm}$ (< 0.05) show an
average occurrence of 22% of the time in Ersa, 20% in Barcelona and only 9.5% in Lampedusa. These features
show that, over the March 2005 – October 2013, POLDER-3 has recorded very low occurrence of pristine days,
i.e. clean conditions associated to low aerosol loads, especially at Lampedusa. As reported in Table 2, the average
daily AOD (865 nm) is 0.09 (standard deviation 0.07) in Ersa, 0.10 (standard deviation 0.04) in Barcelona, and
0.15 (standard deviation 0.18) in Lampedusa, reflecting both higher frequency and intensity of aerosol episodes in
Lampedusa, as illustrated in Figure 7a. This main pattern is also verified for POLDER-3 retrievals of $AOD_c$ and


to a certain extent $AOD_F$, which reach their maximum values in Lampedusa (4.4 and 0.35, respectively). However,
POLDER-3 shows that at 865 nm, the $AOD_F$ is always lower than 0.2 (Figure 7b), except at Lampedusa for a
reduced number of days (4). At this site, peaks of $AOD_F$ seem to be associated to peaks of $AOD_C$, suggesting the
influence of desert dust on both aerosol size components, and/or the double influence of two different aerosol
types (i.e., possibly both dust and anthropogenic). POLDER-3 $AOD_{CS}$ and $AOD_{CNS}$ time series, shown Figure 7d,
are more difficult to interpret, first of all because the sampling is reduced by more than 50% compared to
POLDER-3 retrievals associated to all clear sky pixels (ACSP, i.e., AOD, $AOD_F$, $AOD_C$, AE), due to the necessity
of best viewing conditions (BVC) for their retrieval, as reported in Table 2. Despite this limitation, Figure 7d and
Table 2 show high variability of both spherical and non-spherical aerosols in the coarse mode, with a larger range
of daily values for $AOD_{CNS}$ (up to 1.00 in Lampedusa) than for $AOD_{CS}$ (maximum 0.34 in Barcelona). At the three
sites considered here, POLDER-3 mean retrievals of daily $AOD_{CNS}$ (0.04 – 0.08) are on average more than two
times larger than those of $AOD_{CS}$ (0.02 – 0.03).

**3.4 Inter-annual evolution**

Annual maps of POLDER-3 AOD, $AOD_C$, and $AOD_F$ at 865 nm are displayed for each of the 9 available
observations years (2005 to 2013) in Figure 8. The annual averages are computed over the period March-October
only in order to consistently consider the 9 years in the whole available period. The left period November-February
is hopefully the period where AOD is the lowest in the region (Figure 3). Figure 8 highlights a significant
interannual variation in AOD (left column), characterized by elevated aerosol loads for specific years, as 2007 and
2008, and lower AOD ranges in 2009 and 2013. The interannual variations of POLDER-3 $AOD_C$ (middle column)
tend to be relatively similar to those of AOD, especially over the south part of the basin. Figure 8 also suggests
that the maximum values of $AOD_F$ (right column) were observed in the first half of the period of study, with an
evolution toward more moderate to low loads in fine particles apparent from 2010. Figure S4 of the supplementary
material confirms such an evolution with annual maps of POLDER-3 $AOD_F$ extrapolated at 550 nm for each of
the 9 observation years. The year 2007 appears highly polluted in fine particles over the whole basin. Over the
most eastern part of the region, the intense plume observed by POLDER-3 can be related to the occurrence of
devastating fires in Greece in the summer of 2007, producing large amounts of biomass burning aerosols
transported downwind over the central Mediterranean (Kaskaoutis et al., 2011).
In order to analyze further these interannual evolutions, Figure 9 presents the time series of annual averages (left
column) and monthly anomalies (right column) of POLDER-3 AOD, $AOD_F$, and $AOD_C$ at 865 nm spatially
averaged over the north, central, and south parts of the western Mediterranean basins (defined in Figure 2) for the
period March 2005 – October 2013. The monthly anomalies are computed by subtracting to each monthly averaged
value of a specific year its corresponding long-term monthly average (2005-2013). Linear regressions are applied
to both March-October annual averages and monthly anomalies of POLDER-3 AOD, $AOD_F$, and $AOD_C$ evolution
as a function of time. The values of the slopes, reported in Table 3 provide the sign and magnitude of the trends at
865 nm. Slopes derived from the same analysis of POLDER-3 AOD, $AOD_F$, and $AOD_C$ extrapolated at 550 nm
are reported in Table S1 of supplementary material. The same approach is applied to POLDER-3 AOD, $AOD_F$,
and $AOD_C$ retrievals extracted at Ersa, Barcelona, and Lampedusa. Results are presented Figure 10, with trends
and their statistical level of confidence reported in Table 4 at 865 nm (Table S2 of supplementary material at 550
nm). Overall, this analysis clearly reveals negative values of the trends for all the sub-regions and sites considered


over our study region, highlighting that POLDER-3 has recorded a general decrease of aerosol loads over western
Mediterranean Sea over the period 2005-2013. The decreasing trends recorded for AOD interannual evolution are
found to be statistically significant, at least at the 95% confidence level, over the northern and central part of the
study region and, consistently, at Ersa and Barcelona (top panels of Figure 9 and 10). In contrast, for the
southernmost part of the region (SW MED), the 95% confidence level is not reached when considering annual
means of AOD and when AOD are extracted at Lampedusa (both from annual means and monthly anomalies),
suggesting more uncertainties on the robustness of the decreasing trend. $AOD_C$ interannual evolutions recorded
by POLDER-3 show decreasing trends, although the confidence level of 95% is only reached when considering
monthly anomalies at Barcelona and for the three sub-regions (middle panels of Figure 9 and 10). The absolute
values of the POLDER-3 $AOD_C$ decreasing trends, especially in the northern part of the basin (NW MED, trend $\leq$
$- 0.0012$ yr$^{-1}$) suggest a moderate-to-low decreasing tendency, around -0.01 per decade. Interestingly, POLDER-3
$AOD_F$ interannual evolutions for the three sub-regions (bottom panels of Figure 9) clearly reveal robust decreasing
trends, statistically significant at 99% level (Student's test). As reported in Table 3, considering the northern and
central parts of the study region, $AOD_F$ decreased by $- 0.0020$ yr$^{-1}$ at 865 nm ($- 0.005$ yr$^{-1}$ at 550 nm, Table S1)
whereas the decrease found in the southern part is slightly lower, $- 0.0016$ yr$^{-1}$ at 865 nm ($\leq - 0.004$ yr$^{-1}$ at 550 nm,
Table S1). Analysis of POLDER-3 $AOD_F$ interannual variability at Ersa, Barcelona, and Lampedusa confirm these
downward evolutions, with decreasing trends statistically significant at the 99% confidence level (Table 4 and
bottom panels of Figure 10). The decrease trends seem to be more pronounced in Barcelona (between $- 0.0026$ yr$^-$
$^1$ for annual means and $- 0.0029$ yr$^{-1}$ for monthly anomalies, respectively) than in Lampedusa (between -0.0015 yr$^-$
$^1$ and -0.0017 yr$^{-1}$), with intermediate magnitudes at Ersa (between $- 0.0019$ and $- 0.0024$ yr$^{-1}$). Consistently, the
decreasing trends derived from POLDER-3 $AOD_F$ extrapolated at 550 nm very between values around $– 0.007$ yr$^-$
$^1$ at Barcelona, $- 0. 005/ - 0.006$ yr$^{-1}$ in Ersa, and -0.004 yr$^{-1}$ in Lampedusa (Table S2).
The year-to-year variations in the North Atlantic Oscillation (NAO) have been examined in several past studies to
support interpretation of inter-annual changes of north African dust transport either recorded by different satellite
sensors, especially over the Mediterranean in the 1990s and early 2000s decades (Moulin et al., 1997; Antoine et
Nobileau, 2006) or simulated by regional models (Nabat et al, 2020). In the present paper, we investigate the
relationship between winter (December through March) NAO index defined by Hurrell (1995) and interannual
variations of POLDER-3 AOD, $AOD_F$, and $AOD_C$ from 2005 to 2013 over the three western Mediterranean sub-
regions and sites considered in this work. The winter NAO indexes for the 2005–2013 period were obtained from
"The Climate Data Guide: Hurrell North Atlantic Oscillation (NAO) Index (station-based)"
(https://climatedataguide.ucar.edu/climate-data/hurrell-north-atlantic-oscillation-nao-index-station-based). The
annual means of POLDER-3 AOD and $AOD_F$ do not show any statistically significant correlation with the winter
NAO Index, although the correlation coefficients for annual AOD reach 0.51 at Ersa, and 0.66 for CW MED. The
annual averages of $AOD_C$ confirm a link with the NAO for the CW MED region (r=0.70, with 95% confidence
level). At Ersa, we obtain r=0.54 which is not significant. These correlation levels, not observed in the southern
areas of our study region (Lampedusa or SW MED), strongly suggest that the NAO exerts a control on north
African dust transport rather than on their emissions over source-regions. In order to go further, we examine the
relative frequency of desert dust episodes ($f_D$) by selecting the days associated with POLDER-3 $AOD_C$ 865 nm $\geq$
0.10 for the three-sub regions considered in our study. Figure 11 reports the results for the period 2005–2013
(March-October) along with the time series of the winter NAO Index. A significant correlation is confirmed





between NAO Index and $f_D$ for the central part of the western Mediterranean region (blue curve, R=0.76, with
95% confidence level) and to a lesser extent for the northern part of the western Mediterranean region (green
curve, R=0.65, not significant). For the southern part of the region, the correlation is much lower (r=0.43) although
some connection with NAO is apparent at the beginning of the period (2005–2009), the correlation being strongly
degraded by the opposition observed in 2010 between extremely low NAO index (-4.64) and a relatively high $f_D$
value (36%). It is noticeable that Salvador et al. (2014), in their analysis of interannual variations of African dust
outbreaks for years 2001–2011 over the western Mediterranean basin, excluded the year 2010 from their
correlation plots with NAO indexes considering that it was associated to an atypical low value of the NAO index,
most probably governed by anomalous atmospheric patterns. Interestingly, SW MED is the only of our three
regions where POLDER-3 has recorded a significant decreasing trend in $f_D$ of -2% ($\pm$ 1%) per year over the period
2005-2013 (R=0.68, with 95% confident level).
We also consider the relative frequency of occurrence of clean conditions associated to low aerosol loads recorded
by POLDER-3 at 865 nm for the fine fraction (daily $AOD_F$ < 0.05, top panels), the coarse fraction (daily $AOD_C$ <
0.05, middle panels) and the total aerosol (daily AOD ≤ 0. 10, bottom panels), named $f_{CF}$ (Clean Fine), $f_{CC}$ (Clean
Coarse), and $f_{CT}$ (Clean Total), respectively. Figure 12 reports the year-to-year evolutions of $f_{CF}$, $f_{CC}$, and $f_C$ for the
three sub-regions, NW MED, CW MED, SW MED (left column) and Ersa, Barcelona, Lampedusa (right column).
Clearly, POLDER-3 record an increasing trend in the frequency of occurrence of clean conditions for the fine
fraction of AOD, both for the three sub-regions and three sites. The $f_{CF}$ trends vary between +2% per year (SW
MED and Lampedusa), +3% per year (CW MED, NW MED, Ersa) and +4% per year (Barcelona), with confidence
levels of 99% (except for SW MED where only 95% confidence level is reached). In Barcelona, the increase is
spectacular with clean conditions in fine particles occurring less than 60% of the time between 2005 and 2007
(minimum in 2007, with 51% of frequency) and reaching values above 75% in the 2011-2013 years (maximum in
2013, with 85% of frequency). Such an evolution is consistent with decreasing trends in surface $PM_{2.5}$ at
background sites in Spain and Europe reported in the literature over 2002–2010 (Cusack et al., 2012). Pandolfi et
al. (2016) further observed decreasing trends between 2004 and 2014 in northeastern Spain, both at the background
site of Barcelona and at the regional background site of Montseny, and mostly related them to decreases in
industrial emissions and in secondary sulfate and nitrate fine particle concentrations. Regarding the coarse fraction
of AOD, $f_{CC}$ records some significant year-to-year variability but no tendency, except for the SW MED sub-region
where a low, slightly positive trend (< + 1% per year, not significant) is recorded over the period 2005–2013,
suggesting a possible slow evolution toward cleaner conditions for the coarse aerosol fraction in the southern part
of the basin. Considering the total aerosol loads (bottom panels of Figure 12), $f_{CT}$ evolution shows an increasing
trend (between +2 and +3% per year with a 95% confidence level) for the three sub-regions and three sites
considered.
Figure 13 and Figure 14 compare the 2005-2013 (March - October) mean values of $AOD_F$ and $AOD_C$ respectively
with their anomalies for each year of the period. The year-to-year evolution of $AOD_F$ is clearly characterized by
positive anomalies in the first years of the period of study (especially, 2005-2007), and negative anomalies for the
most recent years. The spatial distribution of these anomalies indicate lower than long term means $AOD_F$ over the
eastern part of the region in 2012, and mostly over the northern and western part of the region in 2013. Annual
anomalies of $AOD_C$ illustrated in Figure 14 highlight elevated loads of coarse aerosols for specific years and areas
of the region, as in 2008 in the southeastern part or in 2012 in the western part of the basin. In contrast, 2009





(southeastern part), 2010 (western part), and 2013 (most of the basin) appear to be associated with lower than
long-term means values of AOD$_C$. These POLDER-3 interannual evolutions tend to confirm the association
between increased dust transport during positive NAO phases (+2.1 in 2008, +3.17 in 2012) and reduced dust
export in negative NAO phases (-4.64 in 2010, -1.97 in 2013), as previously identified in our analysis and former
studies using other satellite aerosol dataset over the Mediterranean basin (Moulin et al., 1997; Antoine and
Nabileau, 2006; Papadimas et al., 2008).

**4 Conclusion**
This study provides the first analysis of the spatial and temporal variability of aerosol properties obtained by the
daily aerosol retrievals over the western Mediterranean Sea from the POLDER-3/PARASOL spaceborne sensor
over its entire period of operation (March 2005 – October 2013). With the exception of two early studies based on
METEOSAT (Moulin et al., 1997; 1998) or SeaWiFS (Antoine and Nabileau, 2006), most of the previous efforts
dedicated to satellite-derived AOD variability analysis over the Mediterranean basin were based on MODIS data
set (Papadimas et al., 2008; Floutsi et al., 2016). On the basis of the quality and robustness of the POLDER-3
ocean operational retrievals over the western Mediterranean (Formenti et al., 2018), we investigated the POLDER-
3 AOD due to different aerosol particle size classes (total, fine and coarse components) and particle shapes (coarse
spherical and non-spherical contributions) in terms of spatial patterns and temporal variability.
The POLDER-3 aerosol record confirms the high influence of north African desert dust over the region, with a
marked maximum in AOD, along with its coarse and coarse non-spherical component in the southernmost part of
the region, associated with a decrease in AE, and a seasonal maximum occurring in Spring and Summer. In
contrast, the coarse spherical component of AOD remains relatively homogenously low all year long over the
region (AOD$_{CS}$< 0.05). The POLDER-3 retrievals of the fine component of AOD show less spatial variability
compared to that observed for the coarse fraction, although AOD$_F$ tend to be larger in the eastern part of our region
of study. Seasonal averages reveal difference by a factor of 2, with minimum AOD$_F$ of 0.02 (in winter in the south
part of the region) and maximum of 0.04 (in spring in the north part of the region) at 865 nm (corresponding
respectively to 0.06 and 0.12 for AOD$_F$ at 550 nm). POLDER-3 also detects a persistent area of relatively high
loads of fine particles in the northern part of the Adriatic Sea, characterized by seasonal averages of AOD$_F$ larger
than 0.12 at 550 nm.
At three sites representative of different typical aerosol conditions over the western Mediterranean Sea (namely
Ersa, Barcelona, and Lampedusa), POLDER-3 retrievals at 865 nm indicate averages contributions to total AOD
at 865 nm ranging between 19 and 20% for coarse spherical particles, 26 and 36% for fine particles (maximum at
Ersa), and 44 and 55% for coarse non-spherical particles (maximum at Lampedusa). At Lampedusa, POLDER-3
daily observations record the occurrence of intense or extreme aerosol events (AOD >1 up to 4.7) consistently
with the higher and more direct influence of severe desert dust episodes at this southernmost site. At these three
sites, daily POLDER-3 AOD$_{865\,nm}$ values above 0.3 are associated with low AE and FMF fraction (mean values
below 0.5 and 21%, respectively), as well as a dominance of the non-spherical particle fraction in the coarse mode
(mean values above 71%), typical of the desert dust influence. The background "clean" conditions associated to
very low aerosol loads (POLDER-3 daily AOD$_{865\,nm}$ values below 0.05) occur 22% of the time around Ersa, 20%
around Barcelona and 9.5% around Lampedusa over the POLDER-3 period (2005-2013), highlighting the scarcity
of pristine days in this region, especially in its southern part.



Interannual evolutions of March to October POLDER-3-derived AOD, $AOD_F$ and $AOD_C$ reveal negative trends
over the period 2005-2013, these trends being more pronounced for AOD, and above all $AOD_F$, than for the $AOD_C$
component. Further analysis suggests a link between winter NAO Index and frequency of desert dust episodes
($AOD_C$ at 865 nm greater than 0.10, $f_D$), especially significant in the central part of the western Mediterranean Sea.
The analysis of the evolution of clean conditions considering both coarse and fine aerosol components recorded
by POLDER-3 over the period March 2005 – October 2013 highlights significant positive trends for clean
conditions in terms of fine particles (between +2 and +4% per year) over the region, whereas no tendency is evident
for the year-to-year evolution of clean conditions for coarse particles. It is noticeable that the annual frequency of
occurrence of clean conditions relative to fine particles reaches values above 75-80% in Ersa and Barcelona in the
last part of the POLDER-3 operation time, i.e. over the years 2010-2013. These values are much above those
retrieved in the beginning of POLDER-3 record (< 57% in Barcelona and < 67% in Ersa) over the years 2005-
2007. Thus, POLDER-3 aerosol dataset analysis strongly suggests a significant improvement in air quality for the
fine mode aerosol component over the western Mediterranean region, consistent with decreasing anthropogenic
emissions and surface $PM_{2.5}$ reported in Europe around the same period. The occurrence of clean conditions in
terms of coarse aerosol component, with no significant detected tendency, is overall less frequent over our region
of study, with annual values generally below 50-60%, highlighting the large regional influence of desert dust
transported from north African sources.
In conclusion, the high-resolved long-time series POLDER-3 data set of AOD of different size classes provides
new and independent insights complementing previous climatology analysis and trend, which, to date, are largely
based on the MODIS long-term satellite dataset (e.g. Papadimas et al., 2008; Floutsi et al., 2016). The integration
of the POLDER-3 dataset should be most useful as a complement to climate regional models aerosol analysis
(Nabat et al., 2013; 2020) for better constraining the evolution and impacts of the variety of aerosols present in the
Mediterranean atmosphere.
In addition, based on the series of POLDER missions, the capability of multi-spectral, multi-directional and multi-
polarization observations, associated with new inversion schemes, to retrieve aerosol optical and microphysical
properties has been successfully proved (Dubovik et al., 2019). Growing attention to polarization observations has
resulted in the 3MI instrument that enhances the POLDER concept with more spectral information and a better
spatial resolution (Fougnie et al., 2018).

**Data availability**
POLDER-3 data extraction was performed with the program PARASOLASCII (http://www-loa.univ-
lille1.fr/~ducos/public/parasolascii/). This version is made available from the AERIS Data and Service Center
(http://www.icare.univ-lille1.fr/parasol). Technical details are described at http://www.icare.univ-
lille1.fr/projects_data/parasol/docs/Parasol_Level-2_format_latest.pdf. The definition of the flag index is detailed
at page 18 (parameter: quality of the fit).
**Competing interests**
FD is guest editor for the ACP Special Issue of the Chemistry and Aerosols Mediterranean Experiment (ChArMEx)
(ACP/AMT inter-journal SI)". The remaining authors declare that they have no conflict of interest.
**Special issue statement**



This article is part of the special issue of the Chemistry and Aerosols Mediterranean Experiment (ChArMEx)
(ACP/AMT inter-journal SI)". It is not associated with a conference.
**Acknowledgements**
This work is part of the ChArMEx project supported by CNRS-INSU, ADEME, Météo-France and CEA in the
framework of the multidisciplinary program MISTRALS (Mediterranean Integrated Studies aT Regional And
Local Scales; http://mistrals-home.org/). It has also been supported by the French National Research Agency
(ANR) through the ADRIMED project (contract ANR-11-BS56-0006) and by the French National Program of
Spatial Teledetection (PNTS, http://www.insu.cnrs.fr/pnts, project n°PNTS-2015-03). L. Mbemba Kabuiku was
granted by the French Environment and Energy Management Agency (ADEME) and National Center of Space
Studies (CNES). The French national center for Atmospheric data and services AERIS provided access to the
POLDER-3 data used.
LOA participates in the CaPPA (Chemical and Physical Properties of the Atmosphere) project funded by the
French National Research Agency (ANR) through the PIA (Programme d'Investissement d'Avenir) under contract
ANR-11-LABX-0005-01, the Regional Council "Hauts-de-France" and the European Regional Development
Fund (ERDF). We would like to thank Marc Mallet and Pierre Nabat (CNRM-Toulouse, France) for fruitful
discussions about the results of this paper.

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




| | AOD | | | AE | | | AOD$_F$ | | | AOD$_C$ | | |
|---|---|---|---|---|---|---|---|---|---|---|---|---|
| | North | Central | South | North | Central | South | North | Central | South | North | Central | South |
| Winter (DJF) | 0.062 | 0.064 | 0.074 | 0.950 | 0.792 | 0.723 | 0.025 | 0.022 | 0.021 | 0.037 | 0.042 | 0.058 |
| Spring (MAM) | 0.106 | 0.115 | 0.155 | 1.064 | 0.855 | 0.724 | 0.043 | 0.038 | 0.040 | 0.063 | 0.078 | 0.115 |
| Summer (JJA) | 0.106 | 0.126 | 0.153 | 0.947 | 0.819 | 0.737 | 0.038 | 0.038 | 0.040 | 0.068 | 0.088 | 0.113 |
| Fall (SON) | 0.079 | 0.086 | 0.104 | 0.963 | 0.831 | 0.734 | 0.033 | 0.030 | 0.031 | 0.047 | 0.057 | 0.074 |
| **Annual** | **0.090** | **0.099** | **0.124** | **0.985** | **0.826** | **0.729** | **0.035** | **0.032** | **0.033** | **0.055** | **0.067** | **0.091** |
| | **Fine Mode Fraction %** | | | AOD$_{CNS}$ | | | AOD$_{CS}$ | | | | | |
| | North | Central | South | North | Central | South | North | Central | South | | | |
| Winter (DJF) | 40 | 34 | 30 | 0.033 | 0.034 | 0.048 | 0.013 | 0.016 | 0.018 | | | |
| Spring (MAM) | 42 | 34 | 29 | 0.048 | 0.062 | 0.088 | 0.021 | 0.026 | 0.029 | | | |
| Summer (JJA) | 36 | 31 | 27 | 0.046 | 0.058 | 0.091 | 0.021 | 0.027 | 0.031 | | | |
| Fall (SON) | 40 | 35 | 30 | 0.041 | 0.047 | 0.059 | 0.015 | 0.019 | 0.023 | | | |
| **Annual** | **40** | **33** | **29** | **0.043** | **0.051** | **0.073** | **0.018** | **0.022** | **0.026** | | | |

**Table 1a.** The 8 (winter) or 9-year (March 2005 – October 2013) climatological seasonal averaged values of POLDER-3
advanced aerosol products at 865 nm for the north (NW MED), central (CW MED), and south (SW MED) parts of western
Mediterranean basins (defined in Figure 2). Maximum values are reported in red, minimum, in blue.

| | AOD | | | AOD$_F$ | | | AOD$_C$ | | | Fine Mode Fraction % | | |
|---|---|---|---|---|---|---|---|---|---|---|---|---|
| | North | Central | South | North | Central | South | North | Central | South | North | Central | South |
| Winter (DJF) | 0.099 | 0.093 | 0.106 | 0.069 | 0.059 | 0.058 | 0.030 | 0.035 | 0.049 | 65 | 60 | 56 |
| Spring (MAM) | 0.168 | 0.166 | 0.204 | 0.118 | 0.104 | 0.109 | 0.049 | 0.062 | 0.095 | 70 | 62 | 57 |
| Summer (JJA) | 0.163 | 0.180 | 0.208 | 0.110 | 0.110 | 0.117 | 0.053 | 0.070 | 0.091 | 66 | 61 | 57 |
| Fall (SON) | 0.126 | 0.128 | 0.144 | 0.089 | 0.082 | 0.084 | 0.037 | 0.046 | 0.060 | 66 | 61 | 57 |
| **Annual** | **0.141** | **0.143** | **0.167** | **0.098** | **0.089** | **0.093** | **0.043** | **0.053** | **0.074** | **67** | **61** | **57** |

**Table 1b.** Same as Table 1a for AOD, AOD$_F$, AOD$_C$, and Fine Mode Fraction at 550 nm for the north (NW MED), central
(CW MED), and south (SW MED) parts of western Mediterranean basins (defined in Figure 2). Maximum values are reported
in red, minimum, in blue.






| | Ersa<br>$N_{ACSP}$ = 1242 - $N_{BVC}$ = 556 | | Barcelona<br>$N_{ACSP}$ = 1241 - $N_{BVC}$ = 540 | | Lampedusa<br>$N_{ACSP}$ = 1320 - $N_{BVC}$ = 612 | |
|---|---|---|---|---|---|---|
| | Mean ± SD | Range<br>Min – Max | Mean ± SD | Range<br>Min – Max | Mean ± SD | Range<br>Min – Max |
| $^{ACSP}AOD_{865\ nm}$ | 0.09 ± 0.07 | 0.01 – 0.68 | 0.10 ± 0.04 | 0.01 – 1.05 | 0.15 ± 0.18 | 0.02 – 4.72 |
| $^{ACSP}AOD_{F\ 865\ nm}$ | 0.03 ± 0.03 | <0.01 – 0.16 | 0.04 ± 0.03 | <0.01 – 0.19 | 0.04 ± 0.03 | <0.01 – 0.35 |
| $^{ACSP}AOD_{C\ 865\ nm}$ | 0.06 ± 0.06 | <0.01 – 0.65 | 0.07 ± 0.07 | <0.01 – 0.94 | 0.11 ± 0.16 | <0.01 – 4.37 |
| $^{ACSP}AE_{865-670}$ | 0.94 ± 0.53 | 0.01 – 2.23 | 0.90 ± 0.50 | -0.07 – 2.33 | 0.67 ± 0.42 | 0.00 – 2.24 |
| $^{ACSP}FMF\ (\%)$ | 38 ± 23 | 3 – 100 | 37 ± 22 | 1 – 97 | 28 ± 18 | 3 – 100 |
| $^{BVC}AOD_{CNS\ 865nm}$ | 0.04 ± 0.04 | <0.01 – 0.48 | 0.05 ± 0.05 | <0.01 – 0.42 | 0.08 ± 0.09 | <0.01 – 1.00 |
| $^{BVC}AOD_{CS\ 865nm}$ | 0.02 ± 0.03 | <0.01 – 0.24 | 0.02 ± 0.03 | <0.01 – 0.34 | 0.03 ± 0.03 | <0.01 – 0.33 |


**Table 2.** Statistics of POLDER-3 daily retrievals of AOD, $AOD_F$, $AOD_C$, AE, FMF (Fine Mode Fraction), $AOD_{CS}$, and $AOD_{CNS}$ at three main stations, Ersa, Barcelona, and Lampedusa for the period March 2005 - October 2013. The numbers of POLDER-3 retrievals available at each station for all clear sky pixels (ACSP) and for best viewing conditions (BVC) are reported.







|  | AOD 865 nm | | AOD$_{COARSE}$ 865 nm | | AOD$_{FINE}$ 865 nm | |
|---|---|---|---|---|---|---|
| Trend per year Region | Annual means | Monthly anomalies | Annual means | Monthly anomalies | Annual means | Monthly anomalies |
| NW MED | **- 0.0030 ± 0.0011*** | **- 0.0031 ± 0.0006**** | - 0.0010 ± 0.0009 | **- 0.0012 ± 0.0005*** | **- 0.0020 ± 0.0005**** | **- 0.0019 ± 0.0003**** |
| CW MED | **- 0.0035 ± 0.0010*** | **- 0.0035 ± 0.0007**** | - 0.0015 ± 0.0009 | **- 0.0016 ± 0.0006**** | **- 0.0020 ± 0.0004**** | **- 0.0019 ± 0.0003**** |
| SW MED | - 0.0037 ± 0.0019 | **- 0.0043 ± 0.0012**** | - 0.0021 ± 0.0016 | **- 0.0027 ± 0.0010*** | **- 0.0016 ± 0.0004**** | **- 0.0016 ± 0.0003**** |

**Table 3.** POLDER-3 865 nm AOD, AOD$_{COARSE}$ and AOD$_{FINE}$ trends per year derived from March-October annual means and
monthly mean anomalies over the 2005-2013 period for NW MED, CW MED, SW MED. The corresponding annual evolutions
are shown in Figure 8. Trends (year$^{-1}$) are shown with their standard deviations (± 1 ). Values in bold indicate statistically
significant trends at * 95% confidence level and ** 99% confidence level, as determined by the Student t-test.

|  | AOD 865 nm | | AOD$_{COARSE}$ 865 nm | | AOD$_{FINE}$ 865 nm | |
|---|---|---|---|---|---|---|
| Trend per year Station | Annual means | Monthly anomalies | Annual means | Monthly anomalies | Annual means | Monthly anomalies |
| Ersa | **- 0.0035 ± 0.0014*** | **- 0.0030 ± 0.0008**** | - 0.0012 ± 0.0012 | - 0.0011 ± 0.0008 | **- 0.0024 ± 0.0004**** | **- 0.0019 ± 0.0003**** |
| Barcelona | **- 0.0050 ± 0.0021*** | **- 0.0046 ± 0.0011**** | - 0.0021 ± 0.0017 | **- 0.0020 ± 0.0009*** | **- 0.0029 ± 0.0005**** | **- 0.0026 ± 0.0004**** |
| Lampedusa | - 0.0037 ± 0.0028 | - 0.0025 ± 0.0018 | - 0.0021 ± 0.0026 | - 0.0009 ± 0.0016 | **- 0.0017 ± 0.0003**** | **- 0.0015 ± 0.0004**** |

**Table 4.** POLDER-3 865 nm AOD, AOD$_{COARSE}$ and AOD$_{FINE}$ trends per year derived from March-October annual means and
monthly mean anomalies over the 2005-2013 period for Ersa, Barcelona, and Lampedusa. The corresponding annual evolutions
are shown in Figure 9. Trends (year$^{-1}$) are shown with their standard deviations (± 1 ). Values in bold indicate statistically
significant trends at * 95% confidence level and ** 99% confidence level, as determined by the Student t-test.





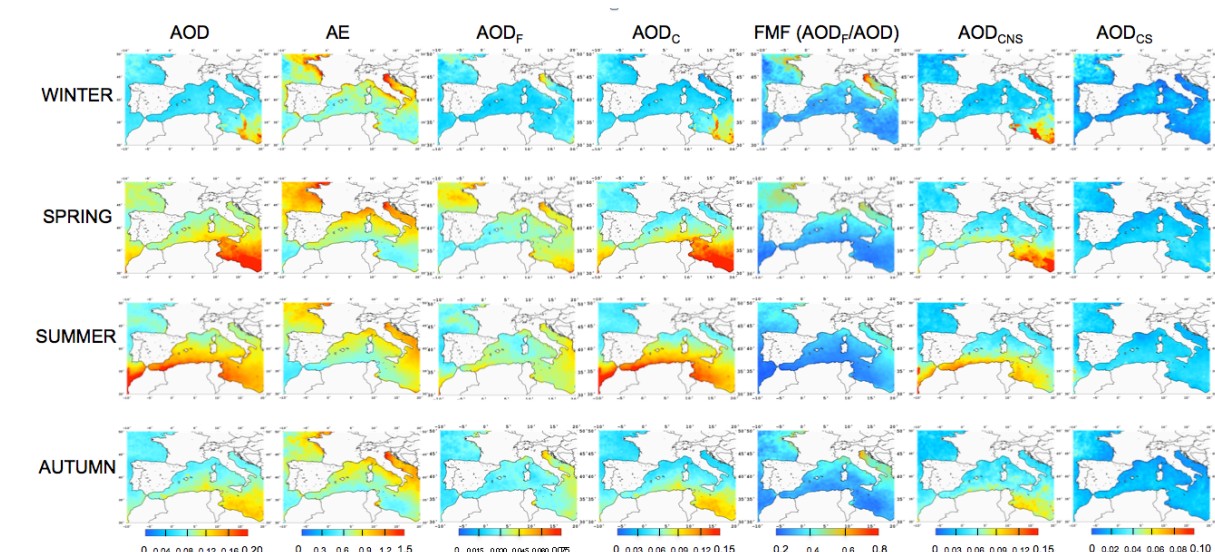

**Figure 1.** Climatological seasonal maps for AOD, AE, $AOD_F$, $AOD_C$, FMF (Fine Mode Fraction derived from $AOD_F$/AOD), $AOD_{CNS}$, and $AOD_{CS}$ retrieved by POLDER-3 at 865 nm over the period March 2005-October 2013. Seasons are ordered from the top to the bottom : Winter is December-January-February, Spring March-April-May, Summer June-July-August, Autumn September-October-November.



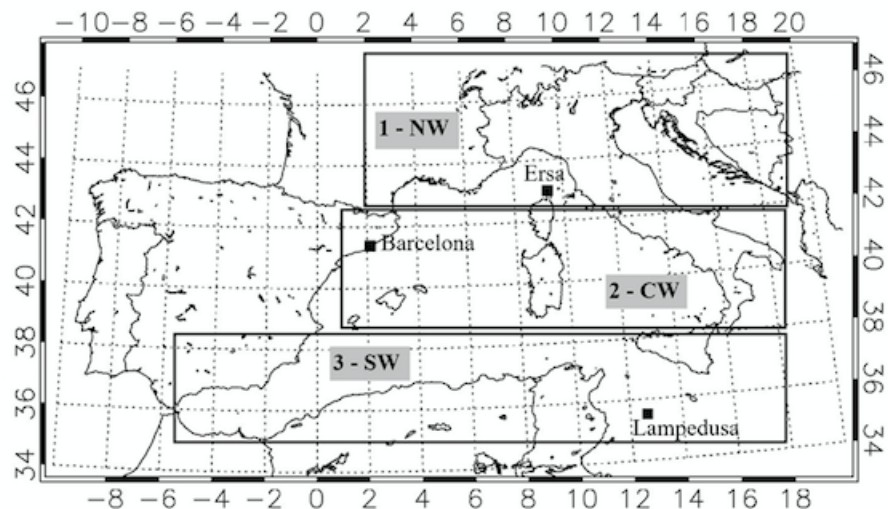

**Figure 2.** Definition of the three geographical sub-regions used to analyze POLDER-3 aerosol retrievals over the area of study: 1. NW Med, 42-46°N, 02°E-20°E – 2. CW MED, 38-42°N, 01°W-20°E, 3. SW MED, 34-38°N, 06°W-20°E. The three sites considered in this study are reported, i.e., Ersa (43.00367°N, 09.35929°E), Barcelona (41.38925°N, 02.11206°E), and Lampedusa (35.51667°N, 12.63167°E).



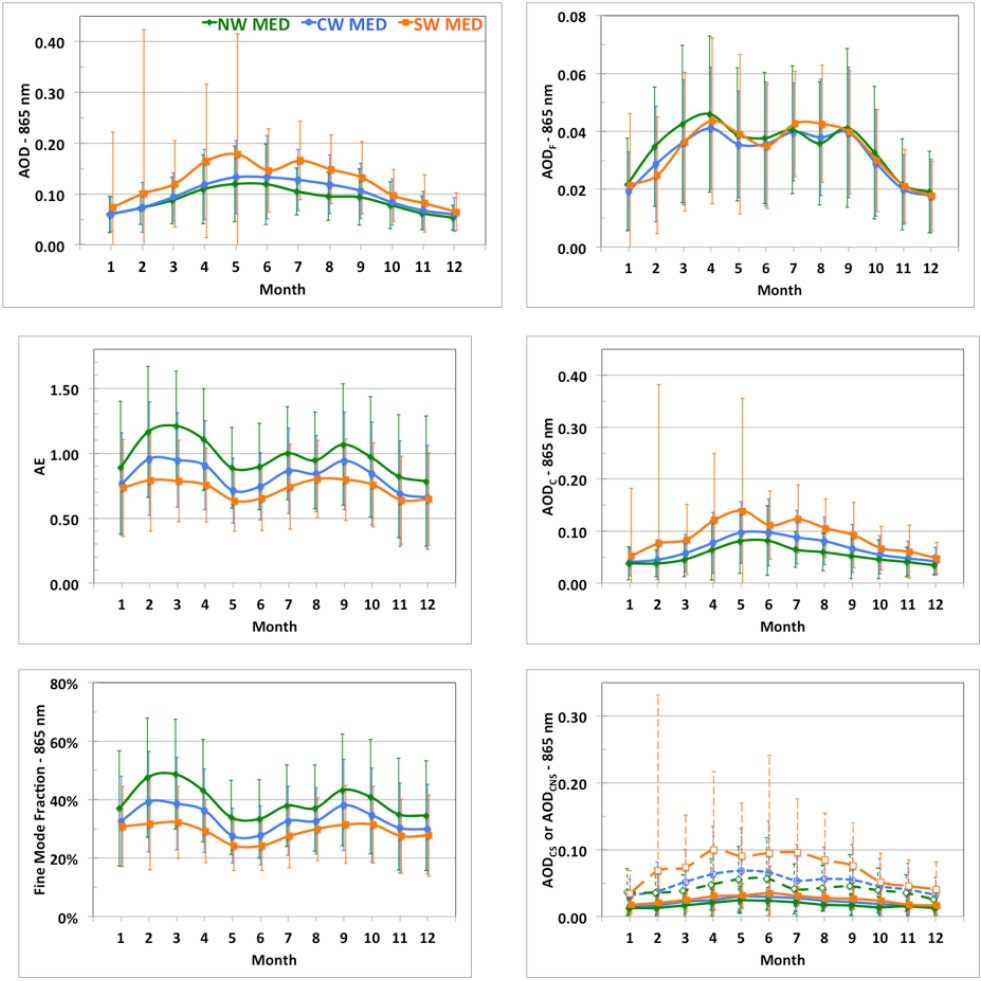

**Figure 3.** The 9-year (March 2005 – October 2013) climatological seasonal cycle of -Left column: AOD (top), Angström Exponent (middle), Fine Mode Fraction (bottom) – Right column: $AOD_{Fine}$ (top), $AOD_{Coarse}$ (middle), $AOD_{Coarse\ Spherical}$ (continuous lines) and $AOD_{Coarse\ Non\ Spherical}$ (dashed lines) (bottom), derived from POLDER-3 at 865 nm. The green, blue, orange curves are respectively for the north (NW MED), central (CW MED), and south (SW MED) parts of western Mediterranean basins (defined in Figure 2).





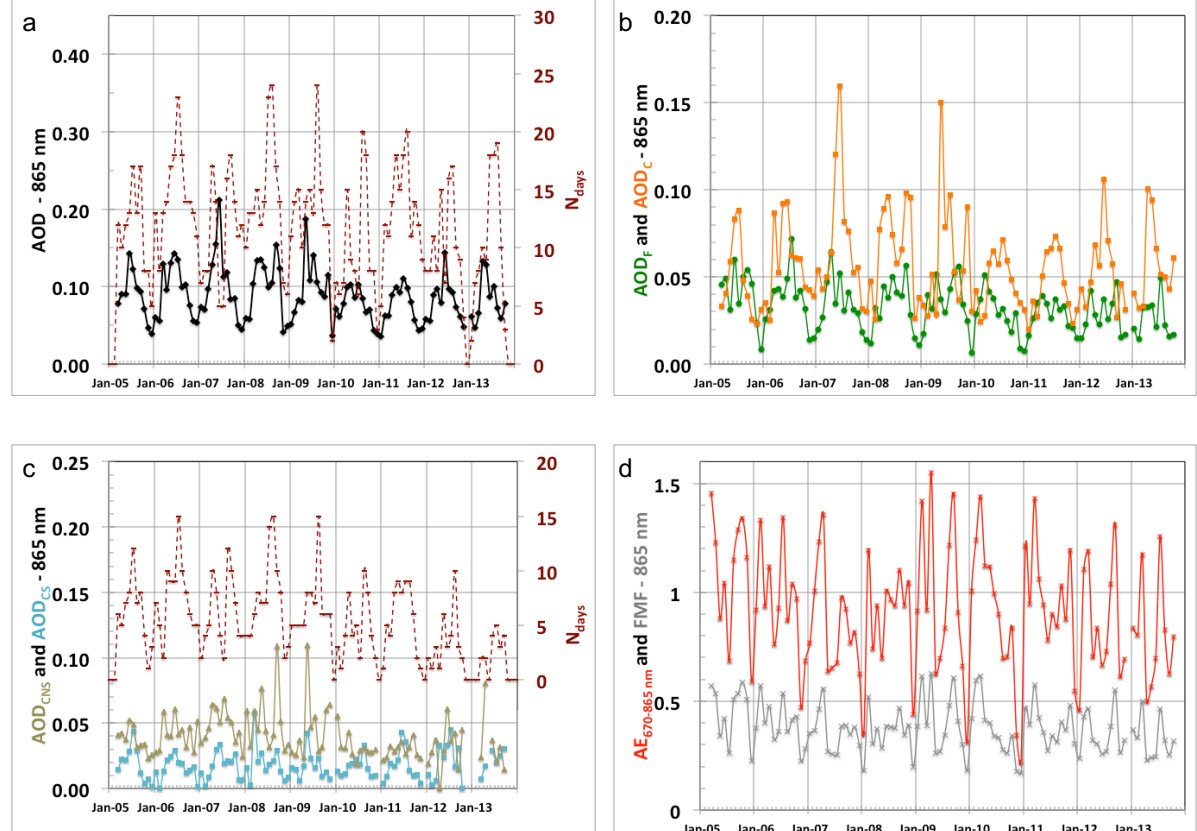

**Figure 4.** POLDER-3 monthly mean retrievals of (a) AOD, (b) $AOD_F$ and $AOD_C$, (c) $AOD_{CNS}$ and $AOD_{CS}$, (d) $AE_{865-670}$ and FMF at 865 nm at Ersa over the period 2005-2013. The number of days of observations available for each month is reported for all clear days (right axis of Figure 4a), and for best viewing conditions (right axis of Figure 4c) necessary for retrievals of $AOD_{CNS}$ and $AOD_{CS}$.





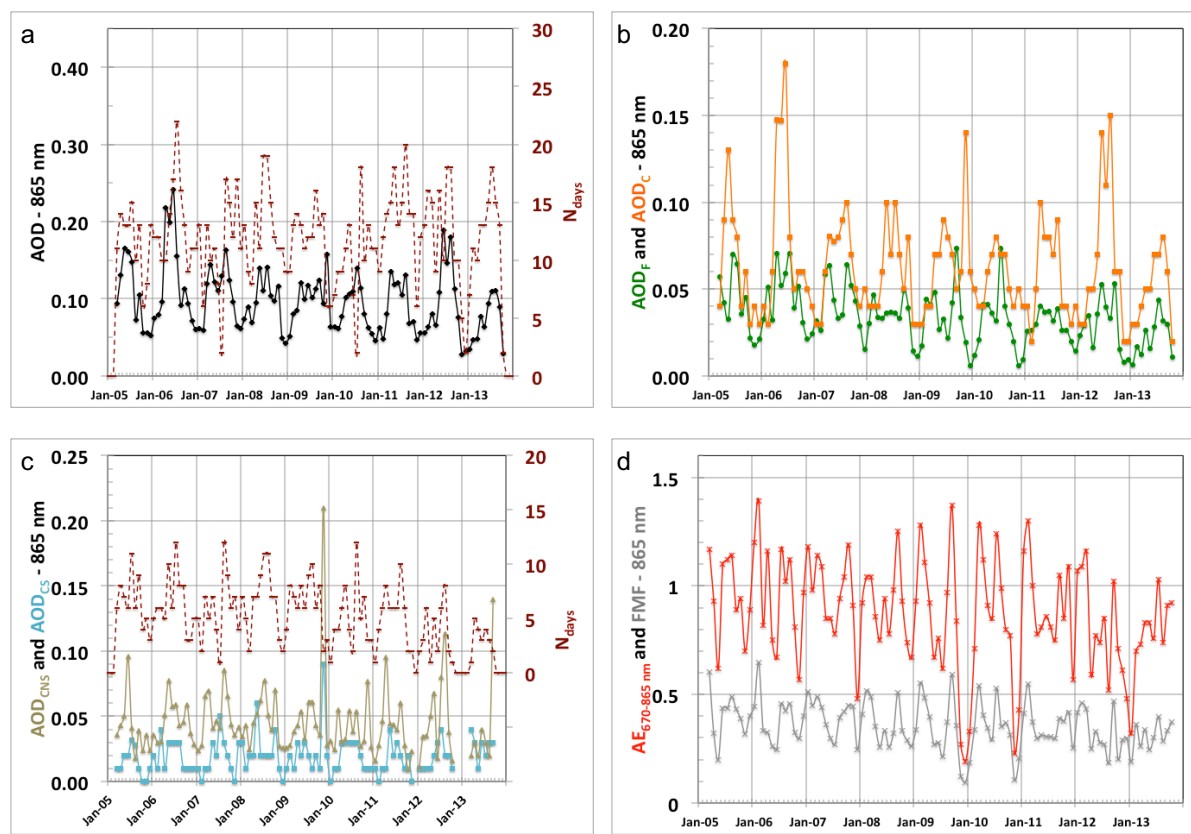

**Figure 5.** Same as Figure 4 for Barcelona.


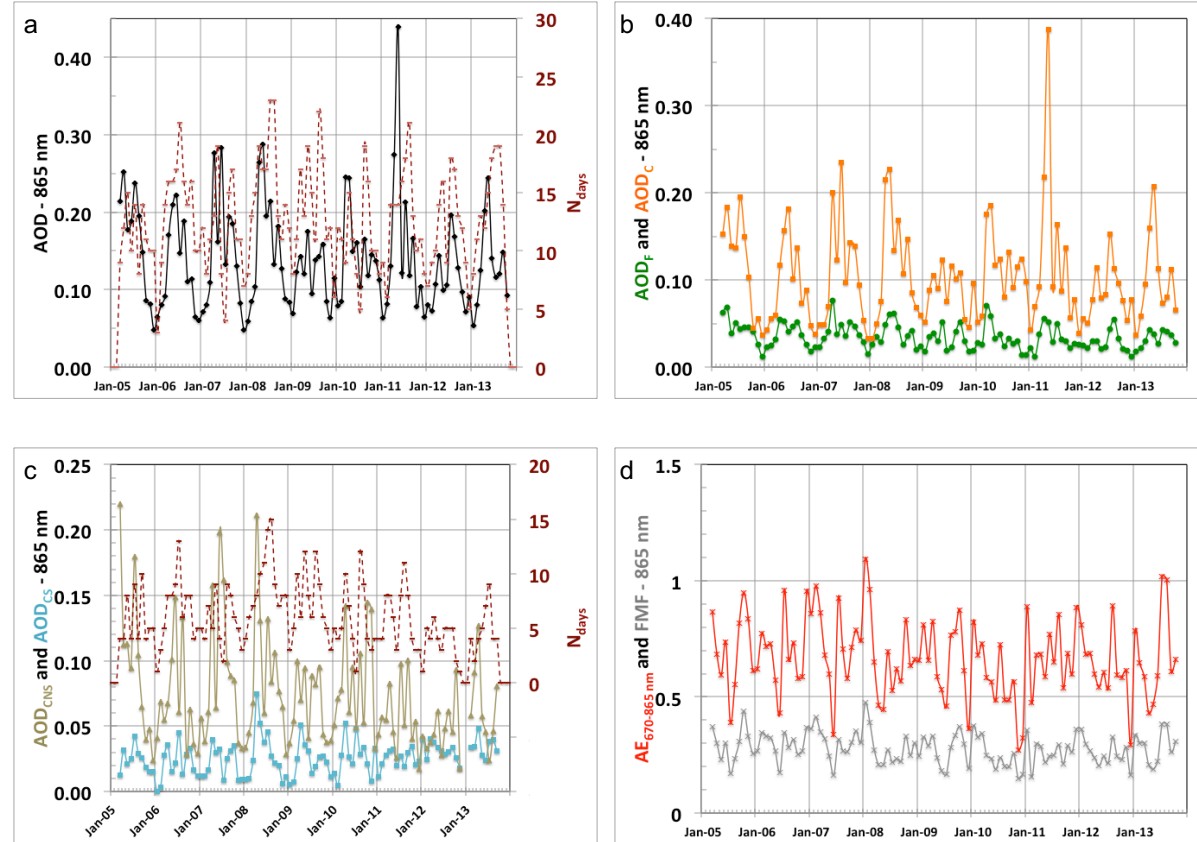

**Figure 6.** Same as Figure 4 for Lampedusa. Note that the scale of Figure 6b is different from that of Figure 4b and 5b.





**Figure 7.** POLDER-3 daily retrievals of a- AOD, b- $AOD_F$, c- $AOD_C$, d- $AOD_{CNS}$ and $AOD_{CS}$ at 865 nm at Ersa (left panels), Barcelona (middle panels), and Lampedusa (right panels) over its whole period of operation (March 4, 2005 – October 10, 2013).





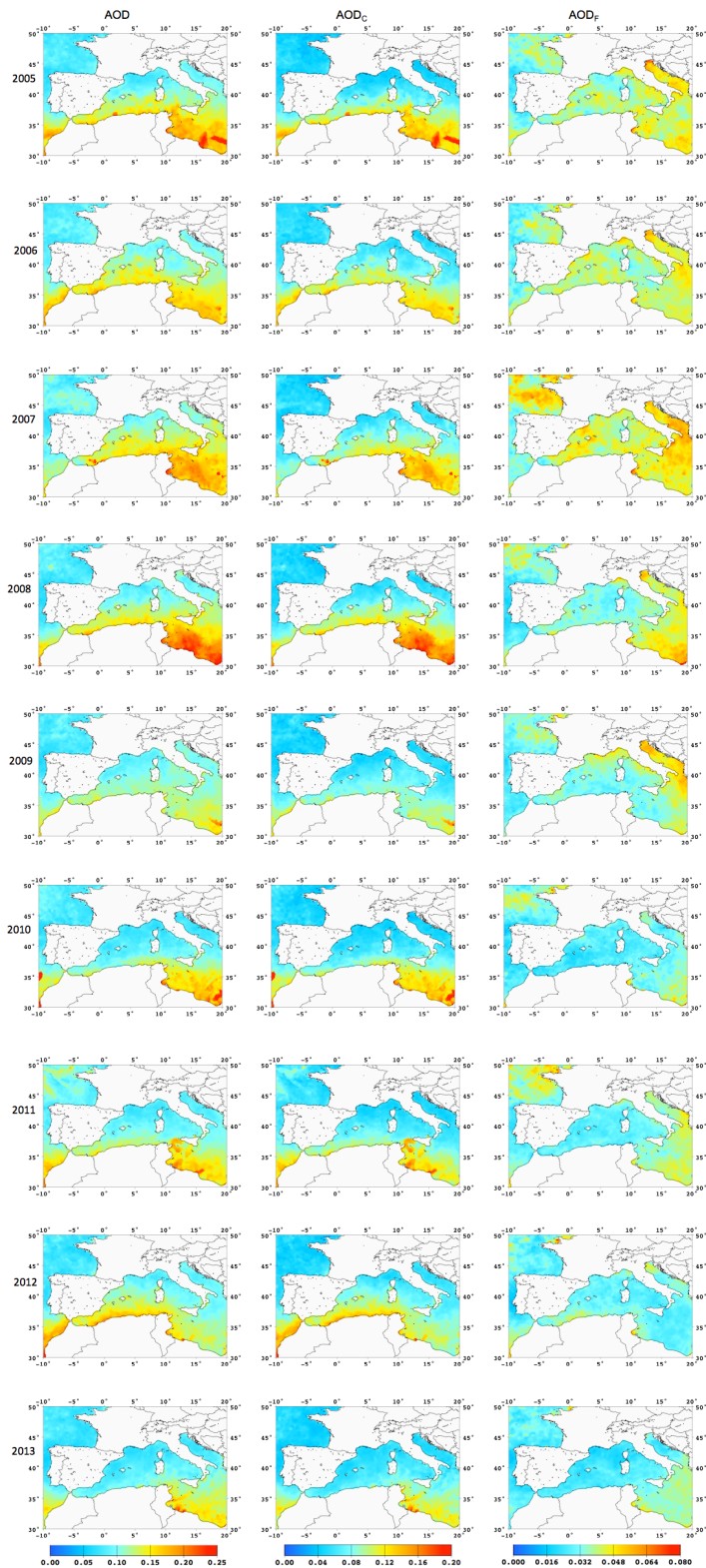

**Figure 8.** March-October annual averages of POLDER-3 AOD (left), AOD$_C$ (middle), AOD$_F$ (right) at 865 nm from 2005 to 2013.





**Figure 9.** March to October yearly means (left column) and monthly anomalies (right column) of POLDER-3 retrievals at 865 nm over the period 2005–2013: AOD (top), AOD$_{COARSE}$ (middle), AOD$_{FINE}$ (bottom) spatially averaged over north (NW MED, green curves), central (CW MED, blue curves), and south (SW MED, orange curves) parts of western Mediterranean basins (defined Figure 2). Trends (year$^{-1}$) are plotted when significant according to the Student t-test, as summarized in Table 3.







**Figure 10.** March to October yearly means (left column) and monthly anomalies (right column) of POLDER-3 retrievals at 865 nm over the period 2005–2013: AOD (top), $AOD_{COARSE}$ (middle), $AOD_{FINE}$ (bottom) extracted at Ersa (pink curves), Barcelona (purple curves), and Lampedusa (brown curves). Trends (year$^{-1}$) are plotted when significant according to the Student t-test, as summarized in Table 4.





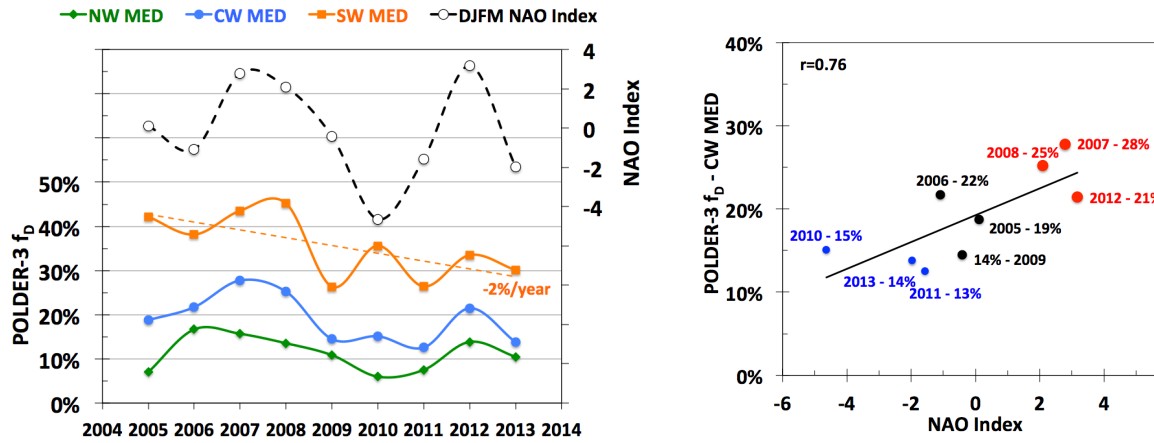

**Figure 11.** Left: Time series of the NAO winter Index (scale on the right axis, open circles) and of the following annual relative frequency ($f_D$) of POLDER-3 $AOD_C$ at 865 nm ≥ 0.10 for the three sub-regions (NW MED in green, CW MED in blue, SW MED in orange) over the period March-2005–October 2013. The only significant trend of $f_D$/year is reported on the graph for SW MED. Right: Scatterplot of $f_D$ versus preceding winter NAO Index for the CW MED region.





**Figure 12.** Left: Time series of annual (March–October) relative frequencies of occurrence of clean conditions for fine mode aerosol component (POLDER-3 $AOD_F$ 865 nm below 0.05, $f_{CF}$; top panel), coarse mode aerosol component (POLDER-3 $AOD_C$ 865 nm below 0.05, $f_{CC}$; middle panel), and total aerosol (POLDER-3 AOD 865 nm lower or equal to 0.10, $f_{CT}$; bottom panel) over the period 2005–2013 for the three sub-regions NW MED, CW MED, SW MED. The dashed lines indicate the multi-year annual averages of relative frequencies. Right: Same for the three sites of Ersa, Barcelona, and Lampedusa.



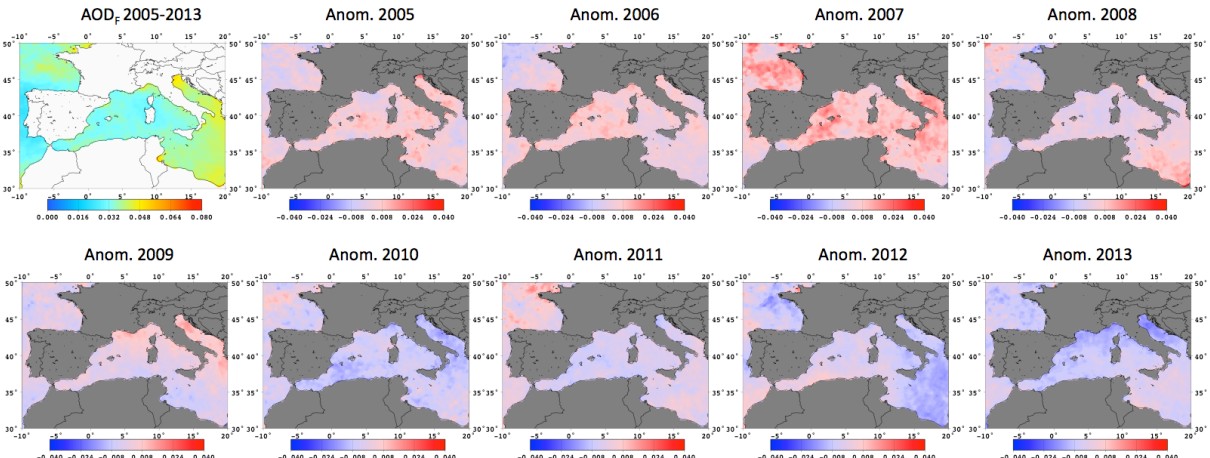

**Figure 13.** POLDER-3 AOD$_F$ at 865 nm averaged over the March-October period and the 9 years 2005-2013 (top left) and associated AOD$_F$ anomalies for each year.



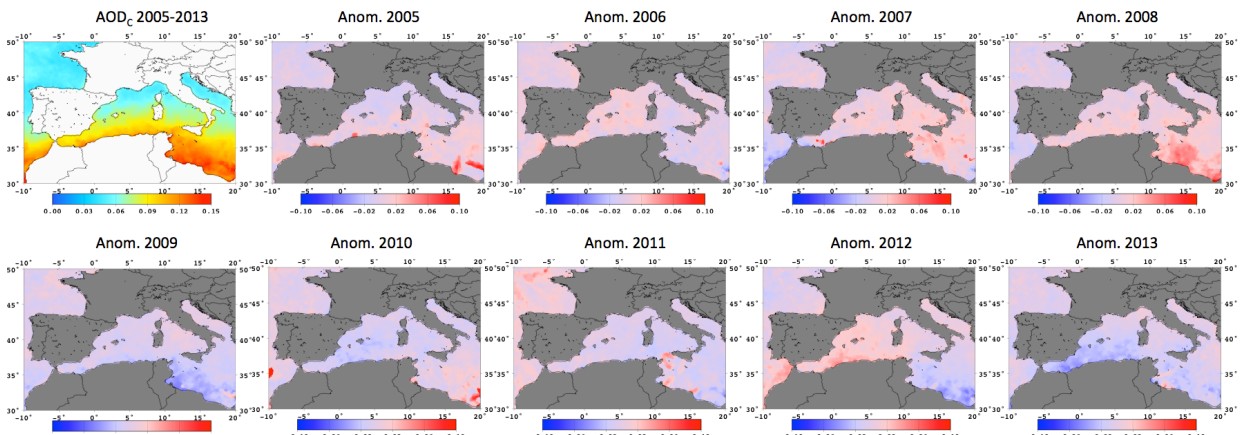

**Figure 14.** POLDER-3 $AOD_C$ at 865 nm averaged over the March-October period and the 9 years 2005-2013 (top left) and associated $AOD_C$ anomalies for each year.