# Peer review of "Aerosol optical properties derived from POLDER-1 3/PARASOL (2005-2013) over the western Mediterranean Sea 2 - Part 2 : Spatial distribution and temporal variability 3"

_Atmospheric Chemistry and Physics, 2020_

## Author Comment (AC1)

**Answers to reviews for acp-2020-1155 – Chiapello et al., Aerosol optical properties derived from POLDER-3/PARASOL (2005-2013) over the western Mediterranean Sea – Part 2: Spatial distribution and temporal variability**

We thank Referee #1 for helpful evaluation of the manuscript and providing us feedback on its content. Detailed responses are presented in the body of the text here below in blue.

**Anonymous Referee #1**

The paper investigates the spatial distribution and temporal evolution of aerosols over the western Mediterranean using the 8.5-year data record of POLDER-3/PARASOL aerosol relevant parameters like AOD, AOD fine and coarse fractions, the ansgtroem exponent (AE), spherical and non-spherical fractions. On top of confirmation of known features (i.e. seasonality and geographical trends) the impact of NAO and AQ trends are also revealed. The paper is well written and constitutes a valuable, yet not unique in terms of new knowledge, contribution to know literature, also supplementing Formenti's paper that focused on the evaluation of the data set. The following comments could be taken into account before publication to ACP.

L 68-78: A brief literature review and basic comparison of the respective conditions met in the eastern part of the Mediterranean basin would be useful to highlight the particularities of this work.

Yes, we agree. Considering as well Referee # 2 comment, we have added a brief literature review, including this topic in the introduction, L 75-87 on the revised manuscript.

L 170-175: The AE exponent has been traditionally used in the past as a proxy for particle size. Now that we are having the fine and coarse mode do you see any difference in the patterns observed? Is there any added value deriving from both types of information to justify their combined use? (a correlation map of AE with AODf and AODc might be interesting in this aspect.

We thank Referee #1 for this question which has been taken into consideration through further analysis of our dataset. Our main conclusion is that POLDER-3 AE is mostly correlated to fine mode fraction (FMF, corresponding to the ratio $AOD_F/AOD$), rather than to $AOD_F$ or $AOD_C$. This result can be clearly observed on Figure 3, considering climatological monthly means of POLDER-3 retrievals. Correlations between AE and FMF reach 0.94, 0.91, and 0.84, respectively for NW MED, CW MED, and SW MED. No correlation is observed between AE and $AOD_C$, although some moderate correlations appear between AE and $AOD_F$ over NW MED (r=0.72) and CW MED (r=0.62). This result is also highlighted, when considering the monthly POLDER-3 retrievals at Ersa (Figure 4), Barcelona (Figure 5), and Lampedusa (Figure 6), with high correlation coefficients between AE and FMF: r=0.96 (Nobs=103, Ersa), r=0.91 (Nobs=104, Barcelona), r=0.92 (Nobs=104, Lampedusa). A sentence has been added in the revised manuscript to clarify this point, L 280-282.

L 185-188: You may also wish to see Hansson et al., (2021) https://doi.org/10.3390/atmos12040445

Thanks for this suggestion: this very recent reference has been added in the revised manuscript, L 197-199

L 215: If only common days are used in Figure 3, would there be any substantial change in the sub-regional comparison?

This point has been verified. No substantial change is evident, as shown by the figures below which compare the sub-regional seasonal cycles obtained when considering all days (continuous lines) and

when selecting only common days (dashed lines), for AOD, $AOD_{Coarse}$, $AOD_{Fine}$ (top figures) and $AOD_{Coarse\ Spherical}$, $AOD_{Coarse\ Non\ Spherical}$ (bottom figures). Minor differences appear for SW MED (orange curves), especially in May (AOD, $AOD_{Coarse}$), and from February to April ($AOD_{Coarse\ Spherical}$ and $AOD_{Coarse\ Non\ Spherical}$). These differences in $AOD_{Coarse\ Spherical}$ and $AOD_{Coarse\ Non\ Spherical}$ are also apparent for CW MED (blue curves, bottom figures), but remain weak.

[Figure]

[Figure]

L 344: In Fig. 7 the y-scale does not allow for discriminating details in the time series. I would suggest an axis break so that the data populated lower part of the plots occupies more surface (like 60% or more).

Thanks for this suggestion. We agree, the quality of Figure 7 was not sufficient in the submitted version. Figure 7 has been completely redone and largely improved with double y-scales to enable analysis of both lower parts and higher parts of the time series. As Referee 2 also mentioned this problem, and as he suggested, these time series have been moved to the supplementary material (new Figure S4 in the revised manuscript), and replaced by more readable histogram frequencies in the main manuscript (modified Figure 7 of the revised manuscript).

L 399: Have you investigated to what extend do the trends in Fig .9 (AODf) might relate, in excess to global brightening, also to the economic crisis and the respective cutting down of many anthropogenic activities in the region? Could the Barcelona case be used as a proxy to support this assumption and further delineate/decompose the trends? (see L 428-432)

This is an interesting point, but it is out of the scope of this article. The Barcelona case is undoubtedly striking. Its use as a proxy in order to relate the observed decreasing $AOD_F$ trend to cutting down of anthropogenic activities would need further analysis including economic data of the region. This aspect has not been investigated yet, as it would need a robust and comprehensive multi-factor statistical analysis that require much more time and effort. However, some references have been included in the text to open the discussion on this worthwile assumption (Querol et al., 2014; Pandolfi et al., 2016). Two sentences have been added in the revised manuscript, L400-404.

---

## Author Comment (AC2)

**Answers to reviews for acp-2020-1155 – Chiapello et al., Aerosol optical properties derived from POLDER-3/PARASOL (2005-2013) over the western Mediterranean Sea – Part 2: Spatial distribution and temporal variability**

We thank Referee #2 for helpful evaluation of the manuscript and providing us feedback on its content. Detailed responses are presented in the body of the text here below in blue.

**Anonymous Referee #2**

This paper describes the spatiotemporal aerosol distribution over the Western Mediterranean (hereafter: WM), based on 8.5-year (2005-2013) POLDER-3/PARASOL aerosol data records. Namely, the total, fine, and coarse AOD at 865 nm, the Angström exponent (670/865 nm), and the spherical and non-spherical contribution of coarse mode AOD is explored. In addition, in order to further support the observed interannual variability of coarse mode AOD, the North Atlantic Oscillation trends have been investigated. The novelty of this work is that the POLDER-3/PARASOL aerosol products is used to study the WM. Having said that, however, the paper mainly confirms already known AOD-relevant patterns in the region.

While the paper is well written and understandable, the authors should definitely revisit the manuscript, try to shorten sections that are too descriptive and summarize better the key findings.

We agree, and have made a lot of effort to reduce the manuscript by removing most of the unnecessary and too descriptive parts. These changes are detailed below. We recall the key findings of the study, which are now better underlined in abstract and conclusion of the revised manuscript:

- We achieve a first regional analysis and interpretation of a unique advanced aerosol satellite data set, POLDER-3, which has provided 8.5-yr of daily accurate and carefully validated retrievals of aerosol optical depth, including its discrimination in size and shape (in the coarse mode) at 865 nm.
- At this wavelength, coarse contribution is high, largely influenced by mineral dust transport over the Western Mediterranean region. Consistently, the non-spherical fraction dominates in the coarse mode, with more homogeneously distributed and relatively low to ranges of coarse spherical AOD.
- Fine aerosols, although contributing moderately to AOD at 865 nm, show increased levels in the east part of the region, especially persistent over the north part of Adriatic Sea. In addition, POLDER-3 reveals a marked decreasing trend of fine AOD over a large part of the Western Mediterranean Sea, strongly suggesting an extensive improvement of air quality in fine particles over the period 2005-2013.
- Inter-annual evolution of coarse aerosols, provided by POLDER-3 coarse AOD appears more influenced by meteorology related to NAO conditions, with less evidence of large-scale significant decreasing.
- In the southern part of the region, POLDER-3 records extreme aerosol events (AOD > 1) related to desert dust transport, and highlights the scarcity of occurrence of clean aerosol conditions (less than 10% of frequency of $AOD_{865nm}$ < 0.05 at Lampedusa).

Overall, our study highlights how the use of advanced aerosol dedicated satellite data sets allows to distinguish long-term evolution of coarse particles, dominated by natural aerosols (desert dust) from that, differently evolving, of mostly anthropogenic fine aerosols. Such a successful resolving is now underlined in Abstract (L 37-40) and Conclusion (L 509-514) of the revised manuscript. It is important, as it open the perspective of applying this kind of satellite investigation to other climate-sensitive regions of the world, that may encounter specific anthropogenic pressure as well as multiple aerosol influences.

Thus, the following comments should be addressed prior to publication to ACP. General comments:

The authors have extrapolated the aerosol relevant parameters from 865 nm to 550 nm, a wavelength much more common in satellite retrievals. However, the discussion e.g., in lines 180-188 (and in general throughout the manuscript) would be more insightful if instead of comparing the known differences of these two wavelengths (e.g. higher AOD at shorter wavelength) the authors provided a comparison with other satellite based retrievals (literature based).

This is an interesting comment, as it is true that the choice of wavelength is of importance. As shown in our analysis of the POLDER-3 retrievals at 865 and 550 nm, 865 nm results is small values of fine-mode AOD and FMF, much reduced compared to 550 nm. We agree that these wavelength differences are known, but we believe it remains important to check the validity of such a spectral behavior on the POLDER-3 retrievals, as satellite retrievals always need to be verified very carefully to ensure robust interpretation. The choice of the wavelength of the POLDER-3 retrievals is a tricky question: 865 nm is one of the wavelengths of the POLDER-3 instrument, thus expected to provide the best accuracy on the retrievals. 550 nm is not, but it is a much more common wavelength for satellite retrievals and aerosol models, including MODIS most widely used data set. POLDER-3 estimates could be less accurate at 550 nm than at 865 nm, due to small errors related to the use of the AE for the conversion. The comparison with other satellite retrievals at 550 nm is also complicated by considering our area of study, limited to the western Mediterranean region, although most of the satellite studies consider the whole Mediterranean basin. As the periods of the published studies generally differ, some bias may also be related to year-to-year changes of aerosol loads. Keeping in mind these limitations, in order to appropriately take into account Referee #2 suggestion, we mainly refer to MODIS-based analysis, often that of Floutsi et al. (2016) in the revised manuscript. Main changes in the text appear L 188 – 189, L 238 – 239, L 497 - 499.

The paper is too descriptive, and unnecessarily long in some parts (e.g. Sec. 3.3.2). Consider shortening some parts, avoid repetitions and draw meaningful comparisons (e.g. with previous studies).

We agree, and have put a lot of effort to reduce some parts of the manuscript and avoid unnecessary descriptions and repetitions. Accordingly, many parts have been shortened and rewritten (abstract, introduction, conclusion). Sections 3.2 and 3.3, including sub-sections 3.3.1 and 3.3.2, have been largely shortened and partly rewritten in the revised manuscript.

In addition, in Section 3.4 (Inter-annual evolution), the number of figures has been reduced, with only March-to-October yearly means shown in the main manuscript (Figure 9 of the revised manuscript), the monthly anomalies being moved to supplementary material (new Figure S6).

We believe the revised paper should be improved now, and thank Referee # 2 for this helpful advice.

Specific comments:

The introduction should be extended, including major findings from previous studies in the study region.

We agree: considering as well a comment from Referee # 1, a substantial part of the introduction has been rewritten and extended to include major finding from previous satellite studies over the region. This new paragraph is shown L 75-87 in the revised manuscript.

Discussion of Fig. S2 (lines 197-213) can be reduced.

Thanks for this suggestion, we fully agree so this has been done in the revised manuscript, L 207-218.

L 218-219: The authors should elaborate more on the two maxima observed

Yes, a sentence has been added in the revised manuscript, including a reference to the literature, L 223-225.

Fig. 7 is very difficult to read at its current form. The authors could consider moving this figure to the supplement, and include another visualization of the daily values in the main manuscript (e.g. histogram).

We fully agree, moreover this is also a comment of Referee #1. Accordingly, Figure 7 (daily time series), has been fully redone and should now be much more readable, with double Y-axis. As suggested, in the revised manuscript it has been moved to the supplementary material and is now Figure S4. New Figure 7 is the representation of the daily time series in form of frequency histograms.

A few typos, grammatical and syntactic mistakes need to be corrected.

Indeed, I have seen a few. Hopefully most of them have been corrected in the revised manuscript.